# A Tractable Inference Perspective of Offline RL

**Xuejie Liu**[1,3*], **Anji Liu**[2*], **Guy Van den Broeck**[2], **Yitao Liang**[1†]
[1]Institute for Artificial Intelligence, Peking University
[2]Computer Science Department, University of California, Los Angeles
[3]School of Intelligence Science and Technology, Peking University
xjliu@stu.pku.edu.cn, liuanji@cs.ucla.edu
guyvdb@cs.ucla.edu, yitaol@pku.edu.cn

## Abstract

A popular paradigm for offline Reinforcement Learning (RL) tasks is to first fit the offline trajectories to a sequence model, and then prompt the model for actions that lead to high expected return. In addition to obtaining accurate sequence models, this paper highlights that *tractability*, the ability to exactly and efficiently answer various probabilistic queries, plays an important role in offline RL. Specifically, due to the fundamental stochasticity from the offline data-collection policies and the environment dynamics, highly non-trivial conditional/constrained generation is required to elicit rewarding actions. While it is still possible to approximate such queries, we observe that such crude estimates undermine the benefits brought by expressive sequence models. To overcome this problem, this paper proposes **Trifle** (**Tr**actable **I**nference for Of**fl**in**e** RL), which leverages modern tractable generative models to bridge the gap between good sequence models and high expected returns at evaluation time. Empirically, Trifle achieves 7 state-of-the-art scores and the highest average scores in 9 Gym-MuJoCo benchmarks against strong baselines. Further, Trifle significantly outperforms prior approaches in stochastic environments and safe RL tasks with minimum algorithmic modifications. [3]

## 1 Introduction

Recent advancements in deep generative models have opened up the possibility of solving offline Reinforcement Learning (RL) [27] tasks with sequence modeling techniques (termed RvS approaches). Specifically, we first fit a sequence model to the trajectories provided in an offline dataset. During evaluation, the model is tasked to sample actions with high expected returns given the current state. Leveraging modern deep generative models such as GPTs [5] and diffusion models [18], RvS algorithms have significantly boosted the performance on various RL problems [1, 6].

Despite its appealing simplicity, it is still unclear whether expressive modeling alone guarantees good performance of RvS algorithms, and if so, on what types of environments. This paper discovers that many common failures of RvS algorithms are not caused by modeling problems. Instead, while useful information is encoded in the model during training, the model is unable to elicit such knowledge during evaluation. Specifically, this issue is reflected in two aspects: (i) *inability to accurately estimate the expected return* of a state and a corresponding action sequence to be executed given near-perfect learned transition dynamics and reward functions; (ii) even when accurate return estimates exist in the offline dataset and are learned by the model, it could still *fail to sample rewarding actions* during evaluation.[4] At the heart of such inferior evaluation-time performance is the fact that highly

---

[*]Equal contribution

[†]Corresponding author

[3]Our code is available at https://github.com/liebenxj/Trifle.git

[4]Both observations are supported by empirical evidence as illustrated in Section 3.

non-trivial conditional generation is required to stimulate high-return actions [32, 3]. Therefore, other than expressiveness, the ability to efficiently and exactly answer various queries (e.g., computing the expected returns), termed *tractability*, plays an equally important role in RvS approaches.

Having observed that the lack of tractability is an essential cause of the underperformance of RvS algorithms, this paper studies *whether we can gain practical benefits from using Tractable Probabilistic Models (TPMs) [35, 7, 23], which by design support exact and efficient computation of certain queries?* We answer the question in its affirmative by showing that we can leverage a class of TPMs that support computing arbitrary marginal probabilities to significantly mitigate the inference-time suboptimality of RvS approaches. The proposed algorithm **Trifle** (**Tr**actable **I**nference for Of**fl**ine RL) has three main contributions:

*Emphasizing the important role of tractable models in offline RL.* This is the first paper that demonstrates the possibility of using TPMs on complex offline RL tasks. The superior empirical performance of Trifle suggests that expressive modeling is not the only aspect that determines the performance of RvS algorithms, and motivates the development of better inference-aware RvS approaches.

*Competitive empirical performance.* Compared against strong offline RL baselines (including RvS, imitation learning, and offline temporal-difference algorithms), Trifle achieves the state-of-the-art result on 7 out of 9 Gym-MuJoCo benchmarks [14] and has the best average score.

*Generalizability to stochastic environments and safe-RL tasks.* Trifle can be extended to tackle stochastic environments as well as safe RL tasks with minimum algorithmic modifications. Specifically, we evaluate Trifle in 2 stochastic OpenAI-Gym [4] environments and action-space-constrained MuJoCo environments, and demonstrate its superior performance against all baselines.

## 2    Preliminaries

**Offline Reinforcement Learning.**    In Reinforcement Learning (RL), an agent interacts with an environment that is defined by a Markov Decision Process (MDP) $\langle \mathcal{S}, \mathcal{A}, \mathcal{R}, \mathcal{P}, d_0 \rangle$ to maximize its cumulative reward. Specifically, the $\mathcal{S}$ is the state space, $\mathcal{A}$ is the action space, $\mathcal{R} : \mathcal{S} \times \mathcal{A} \to \mathbb{R}$ is the reward function, $\mathcal{P} : \mathcal{S} \times \mathcal{A} \to \mathcal{S}$ is the transition dynamics, and $d_0$ is the initial state distribution. Our goal is to learn a policy $\pi(a|s)$ that maximizes the expected return $\mathbb{E}[\sum_{t=0}^{T} \gamma^t r_t]$, where $\gamma \in (0, 1]$ is a discount factor and $T$ is the maximum number of steps.

Offline RL [27] aims to solve RL problems where we cannot freely interact with the environment. Instead, we receive a dataset of trajectories collected using unknown policies. An effective learning paradigm for offline RL is to treat it as a sequence modeling problem (termed RL via Sequence Modeling or RvS methods) [20, 6, 13]. Specifically, we first learn a sequence model on the dataset, and then sample actions conditioned on past states and high future returns. Since the models typically do not encode the entire trajectory, an estimated value or return-to-go (RTG) (i.e., the Monte Carlo estimate of the sum of future rewards) is also included for every state-action pair, allowing the model to estimate the return at any time step.

**Tractable Probabilistic Models.**    Tractable Probabilistic Models (TPMs) are generative models that are designed to efficiently and exactly answer a wide range of probabilistic queries [35, 7, 37]. One example class of TPMs is Hidden Markov Models (HMMs) [36], which support linear time (w.r.t. model size and input size) computation of marginal probabilities and more. Probabilistic Circuits (PCs) [7] are a general class of TPMs. As shown in Figure 1, PCs consist of input nodes ⊙ that represent simple distributions (e.g., Gaussian, Categorical) over one or more variables as well as sum ⊕ and product ⊗ nodes that take other nodes as input and gradually form more complex distributions. Specifically,

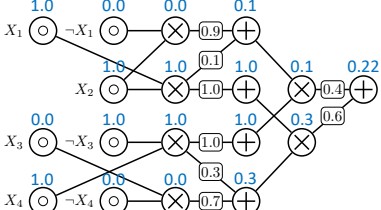

Figure 1: An example PC over boolean variables $X_1, \ldots, X_4$. Every node's probability given input $x_1 x_2 \bar{x}_3 x_4$ is labeled in blue. $p(x_1 x_2 \bar{x}_3 x_4) = 0.22$.

product nodes model factorized distributions over their inputs, and sum nodes build weighted mixtures (mixture weights are labeled on the corresponding edges in Fig. 1) over their input distributions. Please refer to Appx. B for a more detailed introduction to PCs.

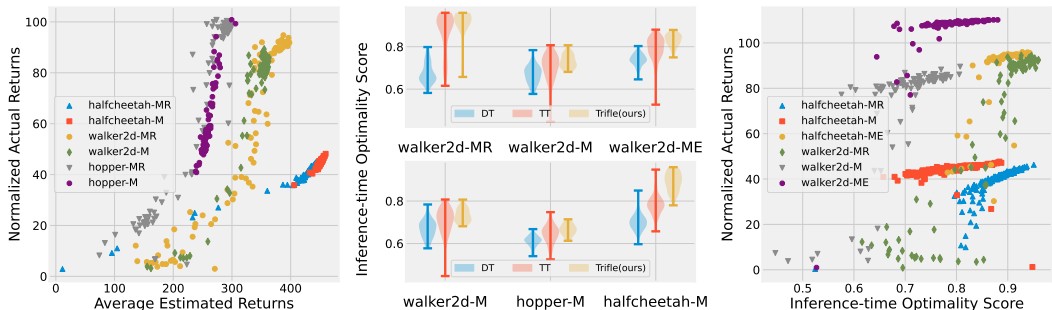

Figure 2: RvS approaches suffer from inference-time suboptimality. **Left:** There is a strong positive correlation between the average estimated returns by Trajectory Transformers (TT) and the actual returns in 6 Gym-MuJoCo environments (MR, M, and ME denote medium-replay, medium, and medium-expert, respectively), which suggests that the sequence model can distinguish rewarding actions from the others. **Middle:** Despite being able to recognize high-return actions, both TT and DT [6] fail to consistently sample such action, leading to bad inference-time optimality; Trifle consistently improves the inference-time optimality score. **Right:** We substantiate the relationship between low inference-time optimality scores and unfavorable environmental outcomes by showing a strong positive correlation between them.

Recent advancements have extensively pushed forward the expressiveness of modern PCs [30, 31, 9], leading to competitive likelihoods on natural image and text datasets compared to even strong Variational Autoencoder [43] and Diffusion model [22] baselines. This paper leverages such advances and explores the benefits brought by PCs in offline RL tasks.

## 3 Tractability Matters in Offline RL

Practical RvS approaches operate in two main phases – training and evaluation. In the training phase, a sequence model is adopted to learn a joint distribution over trajectories of length $T$: $\{(s_t, a_t, r_t, \mathrm{RTG}_t)\}_{t=0}^{T}$.[5] During evaluation, at every time step $t$, the model is tasked to discover an action sequence $a_{t:T} := \{a_\tau\}_{\tau=t}^{T}$ (or just $a_t$) that has high expected return as well as high probability in the prior policy $p(a_{t:T}|s_t)$, which prevents it from generating out-of-distribution actions:

$$p(a_{t:T}|s_t, \mathbb{E}[V_t] \geq v) := \frac{1}{Z} \cdot \begin{cases} p(a_{t:T}|s_t) & \text{if } \mathbb{E}_{V_t \sim p(\cdot|s_t, a_t)}[V_t] \geq v, \\ 0 & \text{otherwise,} \end{cases} \tag{1}$$

where $Z$ is a normalizing constant, $V_t$ is an estimate of the value at time step $t$, and $v$ is a pre-defined scalar chosen to encourage high-return policies. Depending on the problem, $V_t$ could be the labeled RTG from the dataset (e.g., $\mathrm{RTG}_t$) or the sum of future rewards capped with a value estimate (e.g., $\sum_{\tau=t}^{T-1} r_\tau + \mathrm{RTG}_T$) [13, 20].

The above definition naturally reveals two key challenges in RvS approaches: (i) *training-time optimality* (i.e., "expressivity"): how well can we fit the offline trajectories, and (ii) *inference-time optimality*: whether actions can be unbiasedly and efficiently sampled from Equation (1). While extensive breakthroughs have been achieved to improve the training-time optimality [1, 6, 20], it remains unclear whether the non-trivial constrained generation task of Equation (1) hinders inference-time optimality. In the following, we present two general scenarios where existing RvS approaches underperform as a result of suboptimal inference-time performance. We attribute such failures to the fact that these models are limited to answering certain query classes (e.g., autoregressive models can only compute next token probabilities), and explore the potential of *tractable* probabilistic models for offline RL tasks in the following sections.

**Scenario #1** We first consider the case where the labeled RTG belongs to a (near-)optimal policy. In this case, Equation (1) can be simplified to $p(a_t|s_t, \mathbb{E}[V_t] \geq v)$ (choose $V_t := \mathrm{RTG}_t$) since one-step optimality implies multi-step optimality. In practice, although the RTGs are suboptimal, the predicted values often match well with the actual returns achieved by the agent. Take Trajectory

---

[5]To minimize computation cost, we only model truncated trajectories of length $K$ ($K < T$) in practice.

Transformer (TT) [20] as an example, Figure 2 (left) demonstrates a strong positive correlation between its predicted returns (x-axis) and the actual cumulative rewards (y-axis) on six MuJoCo [42] benchmarks, suggesting that the model has learned the "goodness" of most actions. In such cases, the performance of RvS algorithms depends mainly on their inference-time optimality, i.e., whether they can efficiently sample actions with high *predicted* returns. Specifically, let $a_t$ be the action taken by a RvS algorithm at state $s_t$, and $R_t := \mathbb{E}[\text{RTG}_t]$ is the corresponding estimated expected value. We define a proxy of inference-time optimality as the quantile value of $R_t$ in the estimated state-conditioned value distribution $p(V_t|s_t)$.[6] The higher the quantile value, the more frequent the RvS algorithm samples actions with high estimated returns.

We evaluate the inference-time optimality of Decision Transformers (DT) [6] and Trajectory Transformers (TT) [20], two widely used RvS algorithms, on various environments and offline datasets from the Gym-MuJoCo benchmark suite [14]. As shown in Figure 2 (middle), the inference-time optimality is averaged (only) around 0.7 (the maximum possible value is 1.0) for most settings. And these runs with low inference-time optimality scores receive low environment returns (Fig. 2 (right)).

**Scenario #2**  Achieving inference-time optimality becomes even harder when the labeled RTGs are suboptimal (e.g., they come from a random policy). In this case, even estimating the expected future return of an action sequence becomes highly intractable, especially when the transition dynamics of the environment are stochastic. Specifically, to evaluate a state-action pair $(s_t, a_t)$, since $\text{RTG}_t$ is uninformative, we need to resort to the multi-step estimate $V_t^{\text{m}} := \sum_{\tau=t}^{t'-1} r_\tau + \text{RTG}_{t'}$ ($t' > t$), where the actions $a_{t:t'}$ are jointly chosen to maximize the expected return. Take autoregressive models as an example. Since the variables are arranged following the sequential order $\dots, s_t, a_t, r_t, \text{RTG}_t, s_{t+1}, \dots$, we need to explicitly sample $s_{t+1:t'}$ before proceed to compute the rewards and the RTG in $V_t^{\text{m}}$. In stochastic environments, estimating $\mathbb{E}[V_t^{\text{m}}]$ could suffer from high variance as the stochasticity from the intermediate states accumulates over time.

As we shall illustrate in Section 6.2, compared to environments with near-deterministic transition dynamics, estimating the expected returns in stochastic environments using intractable sequence models is hard, and Trifle can significantly mitigate this problem with its ability to marginalize out intermediate states and compute $\mathbb{E}[V_t^{\text{m}}]$ efficiently and exactly.

## 4   Exploiting Tractable Models

The previous section demonstrates that apart from modeling, inference-time suboptimality is another key factor that causes the underperformance of RvS approaches. Given such observations, a natural follow-up question is *whether/how more tractable models can improve the evaluation-time performance in offline RL tasks?* While there are different types of tractabilities (i.e., the ability to compute different types of queries), this paper focuses on studying the additional benefit of *exactly* computing *arbitrary* marginal/condition probabilities. This strikes a proper balance between learning and inference as we can train such a tractable yet expressive model thanks to recent developments in the TPM community [9, 30]. Note that in addition to proposing a competitive RvS algorithm, we aim to highlight the necessity and benefit of using more tractable models for offline RL tasks, and encourage future developments on both inference-aware RvS methods and better TPMs. As a direct response to the two failing scenarios identified in Section 3, we first demonstrate how tractability could help even when the labeled RTGs are (near-)optimal (Sec. 4.1). We then move on to the case where we need to use multi-step return estimates to account for biases in the labeled RTGs (Sec. 4.2).

### 4.1   From the Single-Step Case...

Consider the case where the RTGs are optimal. Recall from Section 3 that our goal is to sample actions from $p(a_t|s_t, \mathbb{E}[V_t] \geq v)$ (where $V_t := \text{RTG}_t$). Prior works use two typical ways to approximately sample from this distribution. The first approach directly trains a model to generate return-conditioned actions: $p(a_t|s_t, \text{RTG}_t)$ [6]. However, since the RTG given a state-action pair is stochastic,[7] sampling

---

[6]Due to the large action space, it is impractical to compute $p(V_t|s_t) := \sum_{a_t} p(V_t|s_t, a_t) \cdot p(a_t|s_t)$. Instead, in the following illustrative experiments, we train an additional GPT model $p(V_t|s_t)$ using the offline dataset.

[7]This is true unless (i) the policy that generates the offline dataset is deterministic, (ii) the transition dynamics is deterministic, and (iii) the reward function is deterministic.

from this RTG-conditioned policy could result in actions with a small probability of getting a high return, but with a low expected return [32, 3].

An alternative approach leverages the ability of sequence models to accurately estimate the expected return (i.e., $\mathbb{E}[\mathrm{RTG}_t]$) of state-action pairs [20]. Specifically, we first sample from a prior distribution $p(a_t|s_t)$, and then reject actions with low expected returns. Such rejection sampling-based methods typically work well when the action space is small (in which we can enumerate all actions) or the dataset contains many high-rewarding trajectories (in which the rejection rate is low). However, the action could be multi-dimensional and the dataset typically contains many more low-return trajectories in practice, rendering the inference-time optimality score low (cf. Fig. 2).

Having examined the pros and cons of existing approaches, we are left with the question of whether a tractable model can improve sampled actions (in this single-step case). We answer it with a mixture of positive and negative results: while computing $p(a_t|s_t, \mathbb{E}[V_t] \geq v)$ is NP-hard even when $p(a_t, V_t|s_t)$ follows a simple Naive Bayes distribution, we can design an approximation algorithm that samples high-return actions with high probability in practice. We start with the negative result.

**Theorem 1.** *Let $a_t := \{a_t^i\}_{i=1}^k$ be a set of $k$ boolean variables and $V_t$ be a categorical variables with two categories $0$ and $1$. For some $s_t$, assume the joint distribution over $a_t$ and $V_t$ conditioned on $s_t$ follows a Naive Bayes distribution: $p(a_t, V_t|s_t) := p(V_t|s_t) \cdot \prod_{i=1}^k p(a_t^i|V_t, s_t)$, where $a_t^i$ denotes the $i^{th}$ variable of $a_t$. Computing any marginal over the random variables is tractable yet conditioning on the expectation $p(a_t|s_t, \mathbb{E}[V_t] \geq v)$ is NP-hard.*

The proof is given in Appx. A. While it seems hard to directly draw samples from $p(a_t|s_t, \mathbb{E}[V_t] \geq v)$, we propose to improve the aforementioned rejection sampling-based method by adding a correction term to the original proposal distribution $p(a_t|s_t)$ to reduce the rejection rate. Specifically, the prior is often represented by an autoregressive model such as GPT: $p_{\mathrm{GPT}}(a_t|s_t) := \prod_{i=1}^k p_{\mathrm{GPT}}(a_t^i|s_t, a_t^{<i})$, where $k$ is the number of action variables and $a_t^i$ is the $i$th variable of $a_t$. We propose to sample every dimension of $a_t$ autoregressively following:

$$\forall i \in \{1, \ldots, k\} \qquad \tilde{p}(a_t^i|s_t, a_t^{<i}; v) := \frac{1}{Z} \cdot p_{\mathrm{GPT}}(a_t^i|s_t, a_t^{<i}) \cdot p_{\mathrm{TPM}}(V_t \geq v|s_t, a_t^{\leq i}), \quad (2)$$

where $Z$ is a normalizing constant and $p_{\mathrm{TPM}}(V_t \geq v|s_t, a_t^{\leq i})$ is a correction term that leverages the ability of the TPM to compute the distribution of $V_t$ given incomplete actions (i.e., evidence on a subset of action variables). Note that while Equation (2) is mathematically identical to $p(a_t|s_t, V_t \geq v)$ when $p = p_{\mathrm{TPM}} = p_{\mathrm{GPT}}$, this formulation gives us the flexibility to use the prior policy (i.e., $p_{\mathrm{GPT}}(a_t^i|s_t, a_t^{<i})$) represented by more expressive autoregressive generative models.

As shown in Figure 2 (middle), compared to using $p(a_t|s_t)$ (as done by TT), the inference-time optimality scores increase significantly when using the distribution specified by Equation (2) (as done by Trifle) across various Gym-MuJoCo benchmarks.

## 4.2  ...To the Multi-Step Case

Recall that when the labeled RTGs are suboptimal, our goal is to sample from $p(a_{t:t'}|s_t, \mathbb{E}[V_t^{\mathrm{m}}] \geq v)$, where $V_t^{\mathrm{m}} := \sum_{\tau=t}^{t'-1} r_\tau + \mathrm{RTG}_{t'}$ is the multi-step value estimate. However, as shown in the second scenario in Section 3, it is hard even to evaluate the expected return of an action sequence due to the inability to marginalize out intermediate states $s_{t+1:t'}$. Empowered by PCs, we can solve this problem by computing the expectation efficiently as it can be broken down into computing conditional probabilities $p(r_\tau|s_t, a_{t:t'})(t \leq \tau < t')$ and $p(\mathrm{RTG}_{t'}|s_t, a_{t:t'})$ (see Appx. C.2 for details):

$$\mathbb{E}[V_t^{\mathrm{m}}] = \sum_{\tau=t}^{t'-1} \mathbb{E}_{r_\tau \sim p(\cdot|s_t, a_{t:t'})}[r_\tau] + \mathbb{E}_{\mathrm{RTG}_{t'} \sim p(\cdot|s_t, a_{t:t'})}[\mathrm{RTG}_{t'}]. \quad (3)$$

We are now left with the same problem discussed in the single-step case – how to sample actions with high expected returns (i.e., $\mathbb{E}[V_t^{\mathrm{m}}]$). Similar to Equation (2), we add correction terms that bias the action (sequence) distribution towards high expected returns. Specifically, we augment the original action probability $\prod_{\tau=t}^{t'} p(a_\tau|s_t, a_{<\tau})$ with terms of the form $p(V_t^{\mathrm{m}} \geq v|s_t, a_{\leq \tau})$ This leads to:

$$\tilde{p}(a_{t:t'}|s_t; v) := \prod_{\tau=t}^{t'} \tilde{p}(a_\tau|s_t, a_{<\tau}; v),$$

Table 1: Normalized Scores on the standard Gym-MuJoCo benchmarks. The results of Trifle are averaged over 12 random seeds (For DT-base and DT-Trifle, we adopt the same number of seeds as [6]). Results of the baselines are acquired from their original papers.

| Dataset | Environment | TT | | TT(+Q) | | DT | | DD | IQL | CQL | %BC | TD3(+BC) |
|---|---|---|---|---|---|---|---|---|---|---|---|---|
| | | base | Trifle | base | Trifle | base | Trifle | | | | | |
| Med-Expert | HalfCheetah | $95.0_{\pm0.2}$ | $\mathbf{95.1}_{\pm0.3}$ | $82.3_{\pm6.1}$ | $\mathbf{89.9}_{\pm4.6}$ | $86.8_{\pm1.3}$ | $\mathbf{91.9}_{\pm1.9}$ | 90.6 | 86.7 | 91.6 | 92.9 | 90.7 |
| Med-Expert | Hopper | $110.0_{\pm2.7}$ | $\mathbf{113.0}_{\pm0.4}$ | $74.7_{\pm6.3}$ | $\mathbf{78.5}_{\pm6.4}$ | $107.6_{\pm1.8}$ | / | 111.8 | 91.5 | 105.4 | 110.9 | 98.0 |
| Med-Expert | Walker2d | $101.9_{\pm6.8}$ | $\mathbf{109.3}_{\pm0.1}$ | $109.3_{\pm2.3}$ | $\mathbf{109.6}_{\pm0.2}$ | $108.1_{\pm0.2}$ | $\mathbf{108.6}_{\pm0.3}$ | 108.8 | $\underline{109.6}$ | 108.8 | 109.0 | 110.1 |
| Medium | HalfCheetah | $46.9_{\pm0.4}$ | $\mathbf{49.5}_{\pm0.2}$ | $48.7_{\pm0.3}$ | $\mathbf{48.9}_{\pm0.3}$ | $42.6_{\pm0.1}$ | $\mathbf{44.2}_{\pm0.7}$ | 49.1 | 47.4 | 44.0 | 42.5 | 48.3 |
| Medium | Hopper | $61.1_{\pm3.6}$ | $\mathbf{67.1}_{\pm4.3}$ | $55.2_{\pm3.8}$ | $\mathbf{57.8}_{\pm1.9}$ | $67.6_{\pm1.0}$ | / | $\underline{79.3}$ | 66.3 | 58.5 | 56.9 | 59.3 |
| Medium | Walker2d | $79.0_{\pm2.8}$ | $\mathbf{83.1}_{\pm0.8}$ | $82.2_{\pm2.5}$ | $\mathbf{84.7}_{\pm1.9}$ | $74_{\pm1.4}$ | $\mathbf{81.3}_{\pm2.3}$ | 82.5 | 78.3 | 72.5 | 75.0 | 83.7 |
| Med-Replay | HalfCheetah | $41.9_{\pm2.5}$ | $\mathbf{45.0}_{\pm0.3}$ | $48.2_{\pm0.4}$ | $\underline{\mathbf{48.9}}_{\pm0.3}$ | $36.6_{\pm0.8}$ | $\mathbf{39.2}_{\pm0.4}$ | 39.3 | 44.2 | 45.5 | 40.6 | 44.6 |
| Med-Replay | Hopper | $91.5_{\pm3.6}$ | $\mathbf{97.8}_{\pm0.3}$ | $83.4_{\pm5.6}$ | $\mathbf{87.6}_{\pm6.1}$ | $82.7_{\pm7.0}$ | / | $\underline{100.0}$ | 94.7 | 95.0 | 75.9 | 60.9 |
| Med-Replay | Walker2d | $82.6_{\pm6.9}$ | $\mathbf{88.3}_{\pm3.8}$ | $84.6_{\pm4.5}$ | $\underline{\mathbf{90.6}}_{\pm4.2}$ | $66.6_{\pm3.0}$ | $\mathbf{73.5}_{\pm0.1}$ | 75.0 | 73.9 | 77.2 | 62.5 | 81.8 |
| **Average Score** | | 78.9 | **83.1** | 74.3 | 77.4 | 74.7 | / | 81.8 | 77.0 | 77.6 | 74.0 | 75.3 |

where $\tilde{p}(a_\tau|s_t, a_{<\tau}; v) \propto p(a_\tau|s_t, a_{<\tau}) \cdot p(V_t^{\mathrm{m}} \geq v|s_t, a_{\leq\tau})$, $a_{<\tau}$ and $a_{\leq\tau}$ represent $a_{t:\tau-1}$ and $a_{t:\tau}$, respectively.[8] In practice, while we compute $p(V_t^{\mathrm{m}} \geq v|s_t, a_{\leq\tau})$ using the PC, $p(a_\tau|s_t, a_{<\tau}) = \mathbb{E}_{s_{t+1:\tau}}[p(a_\tau|s_{\leq\tau}, a_{<\tau})]$ can either be computed exactly with the TPM or approximated (via Monte Carlo estimation over $s_{t+1:\tau}$) using an autoregressive generative model. In summary, we approximate samples from $p(a_{t:t'}|s_t, \mathbb{E}[V_t] \geq v)$ by first sampling from $\tilde{p}(a_{t:t'}|s_t; v)$, and then rejecting samples whose (predicted) expected return is smaller than $v$.

## 5 Practical Implementation

The previous section has demonstrated how to efficiently sample from the expected-value-conditioned policy (Eq. (1)). Based on this sampling algorithm, this section further introduces the proposed algorithm **Trifle** (**Tr**actable **I**nference for Offl**ine** RL). The high-level idea of Trifle is to obtain good action (sequence) candidates from $p(a_t|s_t, \mathbb{E}[V] \geq v)$, and then use beam search to further single out the most rewarding action. Intuitively, by the definition in Equation (1), the candidates are both rewarding and have relatively high likelihoods in the offline dataset, which ensures the actions are within the offline data distribution and prevents overconfident estimates during beam search.

Beam search maintains a set of $N$ (incomplete) sequences each starting as an empty sequence. For ease of presentation, we assume the current time step is $0$. At every time step $t$, beam search replicates each of the $N$ actions sequences into $\lambda \in \mathbb{Z}^+$ copies and appends an action $a_t$ to every sequence. Specifically, for every partial action sequence $a_{<t}$, we sample an action following $p(a_t|s_0, a_{<t}, \mathbb{E}[V_t] \geq v)$, where $V_t$ can be either the single-step or the multi-step estimate depending on the task. Now that we have $\lambda \cdot N$ trajectories in total, the next step is to evaluate their expected return, which can be computed exactly using the PC (see Sec. 4.2). The $N$-best action sequences are kept and proceed to the next time step. After repeating this procedure for $H$ time steps, we return the best action sequence. The first action in the sequence is used to interact with the environment. Please refer to Appx. C for detailed descriptions of the algorithm.

Another design choice is the threshold value $v$. While it is common to use a fixed high return throughout the episode, we follow [12] and use an adaptive threshold. Specifically, at state $s_t$, we choose $v$ to be the $\epsilon$-quantile value of $p(V_t|s_t)$, which is computed using the PC.

## 6 Experiments

This section takes gradual steps to study whether Trifle can mitigate the inference-time suboptimality problem in different settings. First, in the case where the labeled RTGs are good performance indicators (i.e., the single-step case), we examine whether Trifle can consistently sample more

---

[8]We approximate $p(V_t^{\mathrm{m}} \geq v|s_t, a_{\leq\tau})$ by assuming that the variables $\{r_t, \ldots, r_{t'-1}, \mathrm{RTG}_{t'}\}$ are independent. Specifically, we first compute $\{p(r_\tau|s_t, a_{\leq\tau})\}_{\tau=t}^{t'-1}$ and $p(\mathrm{RTG}_{t'}|s_t, a_{\leq\tau})$, and then sum up the random variables assuming that they are independent. This introduces no error for deterministic environments and remains a decent approximation for stochastic environments.

rewarding actions (Sec. 6.1). Next, we further challenge Trifle in highly stochastic environments, where existing RvS algorithms fail catastrophically due to the failure to account for the environmental randomness (Sec. 6.2). Finally, we demonstrate that Trifle can be directly applied to safe RL tasks (with action constraints) by effectively conditioning on the constraints (Sec. 6.3). Collectively, this section highlights the potential of TPMs on offline RL tasks.

## 6.1 Comparison to the State of the Art

As demonstrated in Section 3 and Figure 2, although the labeled RTGs in the Gym-MuJoCo [14] benchmarks are accurate enough to reflect the actual environmental return, existing RvS algorithms fail to effectively sample such actions due to their large and multi-dimensional action space. Figure 2 (middle) has demonstrated that Trifle achieves better inference-time optimality. This section further examines whether higher inference-time optimality scores could consistently lead to better performance when building Trifle on top of different RvS algorithms, i.e., combining $p_{\text{TPM}}$ (cf. Eq. (2)) with different prior policies $p_{\text{GPT}}$ trained by the corresponding RvS algorithm.

**Environment setup**  The Gym-MuJoCo benchmark suite collects trajectories in 3 locomotion environments (HalfCheetah, Hopper, Walker2D) and constructs 3 datasets (Medium-Expert, Medium, Medium-Replay) for every environment, which results in $3 \times 3 = 9$ tasks. For every environment, the main difference between the datasets is the quality of its trajectories. Specifically, the dataset "Medium" records 1 million steps collected from a Soft Actor-Critic (SAC) [16] agent. The "Medium-Replay" dataset adopts all samples in the replay buffer recorded during the training process of the SAC agent. The "Medium-Expert" dataset mixes 1 million steps of expert demonstrations and 1 million suboptimal steps generated by a partially trained SAC policy or a random policy. The results are normalized such that a well-trained SAC model hits 100 and a random policy has a 0 score.

**Baselines**  We build Trifle on top of three effective RvS algorithms: Decision Transformer (DT) [6], Trajectory Transformer (TT) [20] as well as its variant TT(+Q) where the RTGs estimated by summing up future rewards in the trajectory are replaced by the Q-values generated by a well-trained IQL agent [24]. In addition to the above base models, we also compare Trifle against many other strong baselines: (i) Decision Diffuser (DD) [1], which is also a competitive RvS method; (ii) Offline TD learning methods IQL [24] and CQL [26]; (iii) Imitation learning methods like the variant of BC [34] which only uses 10% of trajectories with the highest return, and TD3(+BC) [15].

Since the labeled RTGs are informative enough about the "goodness" of actions, we implement Trifle by adopting the single-step value estimate following Section 4.1, where we replace $p_{\text{GPT}}$ with the policy of the three adopted base methods, i.e., $p_{\text{TT}}(a_t|s_t)$, $p_{\text{TT(+Q)}}(a_t|s_t)$ and $p_{\text{DT}}(a_t|s_t)$.

**Empirical Insights**  Results are shown in Table 1.[9] First, to examine the benefit brought by TPMs, we compare Trifle with three base policies, as the main algorithmic difference is the use of the improved proposal distribution (Eq. (2)) for sampling actions. We can see that Trifle not only achieves a large performance gain over TT and DT in all environments, but also significantly outperforms TT(+Q) where we have access to more accurate labeled values, indicating that Trifle can enhance the inference-time optimality of base policy reliably and benefit from any improvement of the training-time optimality. See Appx. E.1 for more results and ablation studies.

Moreover, compared with all baselines, Trifle achieves the highest average score of 83.1. It also succeeds in achieving 7 state-of-the-art scores out of 9 benchmarks. We conduct further ablation studies on the rejection sampling component and the adaptive thresholding component (i.e., selecting $v$) in Appx. F.

## 6.2 Evaluating Trifle in Stochastic Environments

This section further challenges Trifle on stochastic environments with highly suboptimal trajectories as well as labeled RTGs in the offline dataset. As demonstrated in Section 3, in this case, it is even

---

[9]When implementing DT-Trifle, we have to modify the output layer of DT to make it combinable with TPM. Specifically, the original DT directly predicts deterministic action while the modified DT outputs categorical action distributions like TT. In the 3 unreported hopper environments, the modified DT fails to achieve the original DT scores.

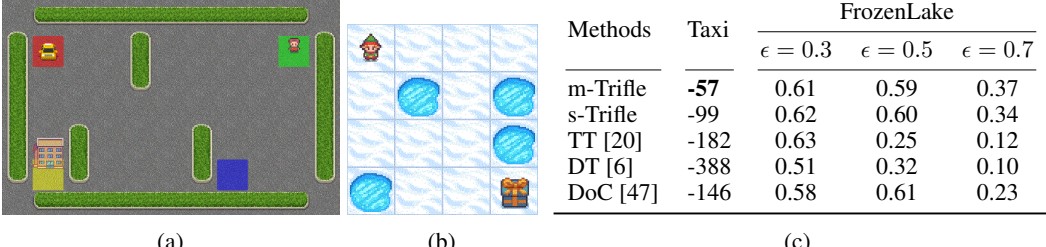

| Methods | Taxi | FrozenLake | | |
|---|---|---|---|---|
| | | $\epsilon = 0.3$ | $\epsilon = 0.5$ | $\epsilon = 0.7$ |
| m-Trifle | **-57** | 0.61 | 0.59 | 0.37 |
| s-Trifle | -99 | 0.62 | 0.60 | 0.34 |
| TT [20] | -182 | 0.63 | 0.25 | 0.12 |
| DT [6] | -388 | 0.51 | 0.32 | 0.10 |
| DoC [47] | -146 | 0.58 | 0.61 | 0.23 |

(a)  (b)  (c)

Figure 3: (a) Stochastic Taxi environment; (b) Stochastic FrozenLake Environment; (c) Average returns on the stochastic environment. All the reported numbers are averaged over 1000 trials.

hard to obtain accurate value estimates due to the stochasticity of transition dynamics.Section 4.2 demonstrates the potential of Trifle to more reliably estimate and sample action sequences under suboptimal labeled RTGs and stochastic environments. This section examines this claim by comparing the five following algorithms:

(i) Trifle that adopts $V_t = \mathrm{RTG}_t$ (termed single-step Trifle or **s-Trifle**); (ii) Trifle equipped with $V_t = \sum_{\tau=t}^{t'} r_\tau + \mathrm{RTG}_{t'}$ (termed multi-step Trifle or **m-Trifle**; see Appx. E.2 for additional details); (iii) TT [20]; (iv) DT [6] (v) Dichotomy of Control (DoC) [47], an effective framework to deal with highly stochastic environments by designing a mutual information constraint for DT training, which is a representative baseline while orthogonal to our efforts.

We evaluate the above algorithms on two stochastic Gym environments: Taxi and FrozenLake. Here we choose the Taxi benchmark for a detailed analysis of whether and how Trifle could overcome the challenges discussed in Section 4.2. Among the first four algorithms, s-Trifle and DT do not compute the "more accurate" multi-step value, and TT approximates the value by Monte Carlo samples. Therefore, we expect their relative performance to be DT ≈ s-Trifle < TT < m-Trifle.

**Environment setup**  We create a stochastic variant of the Gym-Taxi Environment [11]. As shown in Figure 3a, a taxi resides in a grid world consisting of a passenger and a destination. The taxi is tasked to first navigate to the passenger's position and pick them up, and then drop them off at the destination.There are 6 discrete actions available at every step: (i) 4 navigation actions (North, South, East, or West), (ii) Pick-up, (iii) Drop-off. Whenever the agent attempts to execute a navigation action, it has 0.3 *probability of moving toward a randomly selected unintended direction*. At the beginning of every episode, the location of the taxi, the passenger, and the destination are randomly initialized randomly. The reward function is defined as follows: (i) -1 for each action undertaken; (ii) an additional +20 for successful passenger delivery; (iii) -4 for hitting the walls; (iv) -5 for hitting the boundaries; (v) -10 for executing Pick-up or Drop-off actions unlawfully (e.g., executing Drop-off when the passenger is not in the taxi).

Following the Gym-MuJoCo benchmarks, we collect offline trajectories by running a Q-learning agent [45] in the Taxi environment and recording the first 1000 trajectories that drop off the passenger successfully, which achieves an average return of -128.

**Empirical Insights**  We first examine the accuracy of estimated returns for s-Trifle, m-Trifle, and TT. DT is excluded since it does not explicitly estimate the value of action sequences. Figure 4 illustrates the correlation between predicted and ground-truth returns of the three methods. First, s-Trifle performs the worst since it merely uses the inaccurate $\mathrm{RTG}_t$ to approximate the ground-truth return. Next, thanks to its ability to exactly compute the multi-step value estimates, m-Trifle outperforms TT, which approximates the multi-step value with Monte Carlo samples.

We proceed to evaluate their performance in the stochastic Taxi environment. As shown in Figure 3c, the relative performance of the first four algorithms is DT < TT < s-Trifle < m-Trifle, which largely aligns with the anticipated results. The only "surprising" result is the superior performance of s-Trifle compared to TT. One plausible explanation for this behavior is that while TT can better estimate the given actions, it fails to efficiently sample rewarding actions.

Notably, Trifle also significantly outperforms the strong baseline DoC, demonstrating its potential in handling stochastic transitions. To verify this, we further evaluate Trifle on the stochastic FrozenLake

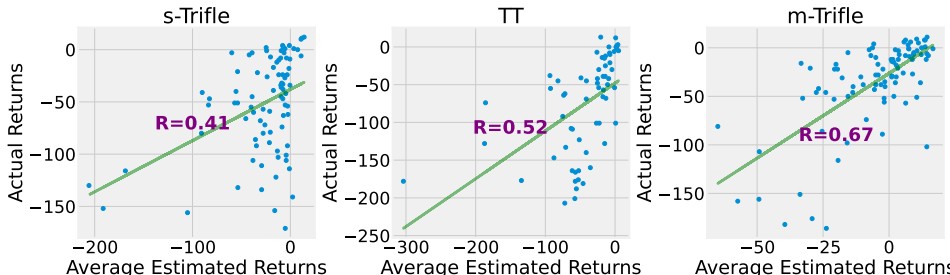

Figure 4: Correlation between average estimated returns and true environmental returns for s-Trifle (w/ single-step value estimates), TT, and m-Trifle (w/ multi-step value estimates) in the stochastic Taxi domain. $R$ denotes the correlation coefficient. The results demonstrate that (i) multi-step value estimates (TT and m-Trifle) are better than single-step estimates (s-Trifle), and (ii) exactly computed multi-step estimates (m-Trifle) are better than approximated ones (TT) in stochastic environments.

environment. Apart from fixing the stochasticity level $p = \frac{1}{3}$,[10] the experiment design follows the DoC paper [47]. For data collection, we perturb the policy of a well-trained DQN (with an average return of 0.7) with the $\epsilon$-greedy strategy. Here $\epsilon$ is a proxy of offline dataset quality and varies from 0.3 to 0.7. As shown in Figure 3c, when the offline dataset contains many successful trials ($\epsilon = 0.3$), all methods perform closely to the optimal policy. As the rollout policy becomes more suboptimal (with the increase of $\epsilon$), the performances of DT and TT drop quickly, while Trifle still works robustly and outperforms all baselines.

## 6.3 Action-Space-Constrained Gym-MuJoCo Variants

This section demonstrates that Trifle can be readily extended to safe RL tasks by leveraging TPM's ability to compute conditional probabilities. Specifically, besides achieving high expected returns, safe RL tasks require additional constraints on the action or states to be satisfied. Therefore, define the constraint as $c$, our goal is to sample actions from $p(a_t|s_t, \mathbb{E}[V_t] \geq v, c)$, which can be achieved by conditioning on $c$ in the candidate action sampling process.

Table 2: Normalized Scores on the Action-Space-Constrained Gym-MuJoCo Variants. The results of Trifle and TT are both averaged over 12 random seeds, with mean and standard deviations reported.

| Dataset | Environment | Trifle | TT |
|---------|-------------|--------|-----|
| Med-Expert | Halfcheetah | **81.9**±4.8 | 77.8±5.4 |
| Med-Expert | Hopper | **109.6**±2.4 | 100.0±4.2 |
| Med-Expert | Walker2d | **105.1**±2.3 | 103.6±4.9 |

**Environment setup**  In MuJoCo environments, each dimension of $a_t$ represents the torque applied on a certain rotor of the hinge joints at timestep $t$. We consider action space constraints in the form of "value of the torque applied to the foot rotor $\leq A$", where $A = 0.5$ is a threshold value, for three MuJoCo environments: Halfcheetah, Hopper, and Walker2d. Note that there are multiple foot joints in Halfcheetah and Walker2d, so the constraint is applied to multiple action dimensions.[11] For all settings, we adopt the "Med-Expert" offline dataset as introduced in Section 6.1.

**Empirical Insights**  The key challenge in these action-constrained tasks is the need to account for the constraints applied to other action dimensions when sampling the value of some action variable. For example, autoregressive models cannot take into account constraints added to variable $a_t^{i+1}$ when sampling $a_t^i$. Therefore, while enforcing the action constraint is simple, it remains hard to simultaneously guarantee good performance. As shown in Table 2, owing to its ability to exactly condition on the action constraints, Trifle outperforms TT significantly across all three environments.

---

[10]When the agent takes an action, it has a probability $p$ of moving in the intended direction and probability $0.5(1 - p)$ of slipping to either perpendicular direction.

[11]We only add constraints to the front joints in the Halfcheetah environment since the performance degrades significantly for all methods if the constraint is added to all foot joints.

# 7 Related Work and Conclusion

In offline reinforcement learning tasks, our goal is to utilize a dataset collected by unknown policies to derive an improved policy without further interactions with the environment. Under this paradigm, we wish to generalize beyond naive imitation learning and stitch good parts of the behavior policy. To pursue such capabilities, many recent works frame offline RL tasks as conditional modeling problems that generate actions with high expected returns [6, 1, 12] or its proxies such as immediate rewards [25, 39, 40]. Recent advances in this line of work can be highly credited to the powerful expressivity of modern sequence models, since by accurately fitting past experiences, we can obtain 2 types of information that potentially imply high expected returns: (i) transition dynamics of the environment, which serves as a necessity for planning in model-based fashion [8], (ii) a decent policy prior which acts more reasonably than a random policy to improve from [20].

While prior works on model-based RL (MBRL) also leverage models of the transition dynamics and the reward function [21, 17, 2], RvS approaches focus more on directly modeling the correlation between actions and their end-performance. Specifically, MBRL approaches focus on planning *only* with the environment model. Despite being theoretically appealing, MBRL requires heavy machinery to account for the accumulated errors during rollout [19, 41] and out-of-distribution problems [48, 38]. All these problems add a significant burden on the inference side, which makes MBRL algorithms less appealing in practice. In contrast, while RvS algorithms can mitigate this inference-time burden by directly learning the correlation between actions and returns, the suboptimality of labeled returns could degrade their performance. One potential solution is to combine RvS algorithms with temporal-difference learning to correct errors in the labeled returns [49, 46].

While also aiming to mitigate the problem caused by suboptimal labeled RTGs, our work takes a different route — by leveraging TPMs to mitigate the inference-time computational burden. Specifically, we identified major problems caused by the lack of tractability in the sequence models, and show that with the ability to compute more queries efficiently, we can partially solve both identified problems.

**Limitations.** One major limitation of Trifle is its dependency on expressive TPMs trained on sequential data — if the TPMs are inaccurate, then Trifle will also have inferior performance. Another limitation is that current implementations of PCs are not as efficient as neural network packages, which could slow down the execution of Trifle.

# Acknowledgements

This work was funded in part by the National Science and Technology Major Project (2022ZD0114902), DARPA ANSR program under award FA8750-23-2-0004, the DARPA PTG Program under award HR00112220005, and NSF grant #IIS-1943641.

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

# Supplementary Material

## A  Proof of Theorem 1

To improve the clarity of the proof, we first simplify the notations in Thm. 1: define $\mathbf{X}$ as the boolean action variables $a_t := \{a_t^i\}_{i=1}^k$, and $Y$ as the variable $V_t$, which is a categorical variable with two categories 0 and 1. We can equivalently interpret $Y$ as a boolean variable where the category 0 corresponds to F and 1 corresponds to T. Dropping the condition on $s_t$ everywhere for notation simplicity, we have converted the problem into the following one:

Assume boolean variables $\mathbf{X} := \{X_i\}_{i=1}^k$ and $Y$ follow a Naive Bayes distribution: $p(\boldsymbol{x}, y) := p(y) \cdot \prod_i p(x_i|y)$. We want to prove that computing $p(\boldsymbol{x}|\mathbb{E}[y] \geq v)$, which is defined as follows, is NP-hard.

$$p(\boldsymbol{x}|\mathbb{E}[y] \geq v) := \frac{1}{Z} \begin{cases} p(\boldsymbol{x}) & \text{if } \mathbb{E}_{y \sim p(\cdot|\boldsymbol{x})}[y] \geq v, \\ 0 & \text{otherwise.} \end{cases} \tag{4}$$

By the definition of $Y$ as a categorical variable with two categories 0 and 1, we have

$$\mathbb{E}_{y \sim p(\cdot|\boldsymbol{x})}[y] = p(y = \mathtt{T}|\boldsymbol{x}) \cdot 1 + p(y = \mathtt{F}|\boldsymbol{x}) \cdot 0 = p(y = \mathtt{T}|\boldsymbol{x}).$$

Therefore, we can rewrite $p(\boldsymbol{x}|\mathbb{E}[y] \geq v)$ as

$$p(\boldsymbol{x}|\mathbb{E}[y] \geq v) := \frac{1}{Z} \cdot p(\boldsymbol{x}) \cdot \mathbb{1}[p(y = \mathtt{T}|\boldsymbol{x}) \geq v],$$

where $\mathbb{1}[\cdot]$ is the indicator function. In the following, we show that computing the normalizing constant $Z := \sum_{\boldsymbol{x}} p(\boldsymbol{x}) \cdot \mathbb{1}[p(y = \mathtt{T}|\boldsymbol{x}) \geq v]$ is NP-hard by reduction from the number partition problem, which is a known NP-hard problem. Specifically, for a set of $k$ numbers $n_1, \ldots, n_k$ ($\forall i, n_i \in \mathbb{Z}^+$), the number partition problem aims to decide whether there exists a subset $S \subseteq [k]$ (define $[k] := \{1, \ldots, k\}$) that partition the numbers into two sets with equal sums: $\sum_{i \in S} n_i = \sum_{j \notin S} n_j$.

For every number partition problem $\{n_i\}_{i=1}^k$, we define a corresponding Naive Bayes distribution $p(\boldsymbol{x}, y)$ with the following parameterization: $p(y = \mathtt{T}) = 0.5$ and [12]

$$\forall i \in [k], \ p(x_i = \mathtt{T}|y = \mathtt{T}) = \frac{1 - e^{-n_i}}{e^{n_i} - e^{-n_i}} \ \text{ and } \ p(x_i = \mathtt{T}|y = \mathtt{F}) = e^{n_i} \cdot \frac{1 - e^{-n_i}}{e^{n_i} - e^{-n_i}}.$$

It is easy to verify that the above definitions lead to a valid Naive Bayes distribution. Further, we have

$$\forall i \in [k], \ \log \frac{p(x_i = \mathtt{T}|y = \mathtt{T})}{p(x_i = \mathtt{T}|y = \mathtt{F})} = n_i \ \text{ and } \ \log \frac{p(x_i = \mathtt{F}|y = \mathtt{T})}{p(x_i = \mathtt{F}|y = \mathtt{F})} = -n_i. \tag{5}$$

We pair every partition $S$ in the number partition problem with an instance $\boldsymbol{x}$ such that $\forall i, x_i = \mathtt{T}$ if $i \in S$ and $x_i = \mathtt{F}$ otherwise. Choose $v = 2/3$, the normalizing constant $Z$ can be written as

$$Z = \sum_{\boldsymbol{x} \in \mathsf{val}(\mathbf{X})} p(\boldsymbol{x}) \cdot \mathbb{1}\big[p(y = \mathtt{T}|\boldsymbol{x}) \geq 2/3\big]. \tag{6}$$

Recall the one-to-one correspondence between $S$ and $\boldsymbol{x}$, we rewrite $p(y = \mathtt{T}|\boldsymbol{x})$ with the Bayes formula:

$$\begin{aligned} p(y = \mathtt{T}|\boldsymbol{x}) &= \frac{p(y = \mathtt{T}) \prod_i p(x_i|y = \mathtt{T})}{p(y = \mathtt{T}) \prod_i p(x_i|y = \mathtt{T}) + p(y = \mathtt{F}) \prod_i p(x_i|y = \mathtt{F})}, \\ &= \frac{1}{1 + e^{-\sum_i \log \frac{p(x_i|y=\mathtt{T})}{p(x_i|y=\mathtt{F})}}}, \end{aligned}$$

---

[12]Note that we assume the naive Bayes model is parameterized using log probabilities.

$$= \frac{1}{1 + e^{-(\sum_{i \in S} n_i - \sum_{j \notin S} n_j)}},$$

where the last equation follows from Equation (5). After some simplifications, we have

$$\mathbb{1}\big[p(y = \texttt{T}|\boldsymbol{x}) \geq 2/3\big] = \mathbb{1}\big[\sum_{i \in S} n_i - \sum_{j \notin S} n_j \geq 1\big].$$

Plug back to Equation (6), we have

$$Z = \sum_{S \subseteq [k]} p(\boldsymbol{x}) \cdot \mathbb{1}\big[\sum_{i \in S} n_i - \sum_{j \notin S} n_j \geq 1\big],$$

$$= \frac{1}{2} \sum_{S \subseteq [k]} p(\boldsymbol{x}) \cdot \mathbb{1}\big[\sum_{i \in S} n_i - \sum_{j \notin S} n_j \neq 0\big],$$

where the last equation follows from the fact that (i) if $\boldsymbol{x}$ satisfy $\sum_{i \in S} n_i - \sum_{j \notin S} n_j \geq 1$ then $\bar{\boldsymbol{x}}$ has $\sum_{i \in S} n_i - \sum_{j \notin S} n_j \leq -1$ and vise versa, and (ii) $\sum_{i \in S} n_i - \sum_{j \notin S} n_j$ must be an integer.

Note that for every solution $S$ to the number partition problem, $\sum_{i \in S} n_i - \sum_{j \notin S} n_j = 0$ holds. Therefore, there exists a solution to the defined number partition problem if $Z < \frac{1}{2}$. $\qquad\square$

# B   Introduction to Probabilistic Circuits

Probabilistic circuits (PCs) represent a wide class of TPMs that model probability distributions with a parameterized directed acyclic computation graph (DAG). Specifically, a PC $p(\mathbf{X})$ defines a joint distribution over a set of random variables $\mathbf{X}$ by a single root node $n_r$. A PC contains three kinds of computational nodes: *input* $\odot$, *sum* $\oplus$, and *product* $\otimes$. The example PC in Figure 1 defines a joint distribution over 4 random variables $X_1, X_2, X_3, X_4$. Each leaf node in the DAG serves as an *input* node that encodes a univariate distribution (e.g., Guassian, Categorical), while *sum* nodes or *product* nodes are inner nodes, distinguished by whether they are doing mixture or factorization over their child distributions (denoted $\mathsf{in}(n)$). Formally, PCs define probability distributions in the following recursive way:

$$p_n(\boldsymbol{x}) := \begin{cases} f_n(\boldsymbol{x}) & \text{if } n \text{ is an input unit,} \\ \sum_{c \in \mathsf{in}(n)} \theta_{n,c} \cdot p_c(\boldsymbol{x}) & \text{if } n \text{ is a sum unit,} \\ \prod_{c \in \mathsf{in}(n)} p_c(\boldsymbol{x}) & \text{if } n \text{ is a product unit,} \end{cases} \tag{7}$$

where $\theta_{n,c}$ represents the parameter corresponding to edge $(n, c)$ in the DAG. For sum nodes, we have $\sum_{c \in \mathsf{in}(n)} \theta_{n,c} = 1, \theta_{n,c} \geq 0$, and we assume w.l.o.g. that a PC alternates between the sum and product layers before reaching its inputs.

## B.1   Tractability and Expressivity

The expressivity of PCs comes from the ability to combine simpler distributions with sum and product nodes to form more complex distributions. Therefore, to increase the capacity of a PC, we can add more nodes to its DAG or find a better structure that is more tailored to the target data distribution. On the other hand, tractability, the ability to answer certain probabilistic queries efficiently and exactly, is guaranteed by certain *structural properties* of PCs. For example, with *smoothness* and *decomposability* defined in the following, PCs can compute arbitrary marginal and conditional probability in time linear with respect to its size (number of edges in its DAG).

**Definition 1** (Decomposability). A PC is decomposable if for every product unit $n$, its children have disjoint scopes:

$$\forall c_1, c_2 \in \mathsf{in}(n) \, (c_1 \neq c_2), \; \phi(c_1) \cap \phi(c_2) = \varnothing.$$

**Definition 2** (Smoothness). A PC is smooth if for every sum unit $n$, its children have the same scope:

$$\forall c_1, c_2 \in \mathsf{in}(n), \; \phi(c_1) = \phi(c_2).$$

As a key procedure used in Trifle, we describe how to compute marginal queries given a smooth and decomposable PC. First, we assign probabilities to all input nodes. For an input node defined on variable $X$, if evidence on $X$ is provided in the query, we set the output probability following Equation (7). Otherwise, we set the output probability to 1. Next, we do a feedforward pass over all inner (sum and product) nodes following Equation (7). The final output at the root node is the desired marginal probability.

In addition to marginal and conditional probabilities, PCs can efficiently and exactly compute other queries including maximize a-posterior and various information-theoretic queries given some additional structural constraints. Please refer to [44] for a comprehensive overview.

## B.2    Adopted PC Structure And Parameter Learning Algorithm

For all tasks/offline datasets, we adopt the Hidden Chow-Liu Tree (HCLT) PC structure proposed by [29] as it has been shown to perform well across different data types.

Following the definition in Equation (7), a PC takes as input a sample $\boldsymbol{x}$ and outputs the corresponding probability $p_n(\boldsymbol{x})$. Given a dataset $\mathcal{D}$, the PC optimizer takes the PC parameters (consisting of sum edge parameters and input node/distribution parameters) as input and aims to maximize the MLE objective $\sum_{\boldsymbol{x} \in \mathcal{D}} \log p_n(x)$. Since PCs can be deemed as latent variable models with hierarchically nested latent space [33], the Expectation-Maximization (EM) algorithm is usually the default choice for PC parameter learning. We adopt the full-batch EM algorithm proposed in [35].

Before tuning the parameters with EM, we adopt the latent variable distillation (LVD) technique proposed in [30] to initialize the PC parameters. Specifically, the neural embeddings used for LVD are acquired by a BERT-like Transformer [10] trained with the Masked Language Model task. To acquire the embeddings of a subset of variables $\phi$, we feed the Transformer with all other variables and concatenate the last Transformer layer's output for the variables $\phi$. Please refer to the original paper for more details.

We use the same quantile dataset discretized from the original Gym-MuJoCo dataset as done by TT [20], where each raw continuous variable is divided into 100 categoricals, and each categorical represents an equal amount of probability mass under the empirical data distribution.

## C    Algorithm Details of Trifle

This section provides a detailed description of the algorithmic procedure of Trifle with single-/multi-step value estimates.

### C.1    Trifle with Single-/Multi-Step Value Estimates

Similar to other RvS algorithms, Trifle first trains sequence models given truncated trajectories $\{(s_t, a_t, r_t, \mathrm{RTG}_t)\}_t$. Specifically, we fit two sequence models: an autoregressive Transformer following prior work [20] as well as a PC, where the training details are introduced in Appx. B.2.

During the evaluation phase, at time step $t$, Trifle is tasked to generate $a_t$ given $s_{\leq t}$ and other relevant information (such as rewards collected in past steps). As introduced in Section 5, Trifle generally works in two phases: rejection sampling for action generation and beam search for action selection. The main algorithm is illustrated in Algorithm 1, where we take the current state $s_t$ as well as the past trajectory $\tau_{<t}$ as input, utilize the specified value estimate $f_v$ as a heuristic to guide beam search, and output the best trajectory. Note that $f_v$ is a subroutine of our algorithm that uses the trained sequence models to compute certain quantities, which will be detailed in subsequent parts. After that, we extract the current action $a_t$ from the output trajectory to execute in the environment.

At the first step of the beam search, we perform rejection sampling to obtain a candidate action set $\mathbf{a_t}$ (line 4 of Algorithm 1). The concrete rejection sampling procedure for s-Trifle is detailed in Algorithm 2. The major modification of m-Trifle compared to s-Trifle is the adoption of a multi-step value estimate instead of the single-step value estimate, which is also shown in Algorithm 3. Specifically, Algorithm 3 is used to replace the value function $f_v$ shown in Algorithm 1.

---

**Algorithm 1** Trifle with Beam Search

---

1: **Input:** past trajectory $\tau_{<t}$, current state $s_t$, beam width $N$, beam horizon $H$, scaling ratio $\lambda$, sequence model $\mathcal{M}$, value function $f_v$       $\triangleright$ $f_v = \mathbb{E}[V_t]$ for s-Trifle and $\mathbb{E}[V_t^m]$ for m-Trifle
2: **Output:** The best action $a_t$
3: Let $\mathbf{x_t} \leftarrow \mathtt{concat}(\tau_{<t}, s_t).\mathtt{reshape}(1, -1).\mathtt{repeat}(N, \mathtt{dim} = 0)$     $\triangleright$ Batchify the input trajectory
4: Perform rejection sampling to obtain $\mathbf{a_t}$ using Algorithm 2       $\triangleright$ cf. Algorithm 2
5: Initialize $X_0 = \mathtt{concat}(\mathbf{x_t}, \mathbf{a_t})$
6: **foreach** $t = 1, ..., H$
7:     $X_{t-1} \leftarrow X_{t-1}.\mathtt{repeat}(\lambda, \mathtt{dim} = 0)$       $\triangleright$ Scale the number of trajectories from $N$ to $\lambda N$
8:     $\mathcal{C}_t \leftarrow \{\mathtt{concat}(\mathbf{x}_{t-1}, x) \mid \forall \mathbf{x}_{t-1} \in X_{t-1}, \mathtt{sample}\ x \sim p_{\mathcal{M}}(\cdot \mid \mathbf{x}_{t-1})\}$     $\triangleright$ Candidate next-token prediction
9:     $X_t \leftarrow \mathtt{topk}_{X \in \mathcal{C}_t}\ (f_v(X), \mathtt{k} = N)$       $\triangleright$ keep $N$ most rewarding trajectories
10: **end foreach**
11: $X_m \leftarrow \mathtt{argmax}_{X \in X_H}\ f_v(X)$
12: **return** $a_t$ in $X_m$

---

---

**Algorithm 2** Rejection Sampling with Single-step Value Estimate

---

1: **Input:** past trajectory $\tau_{<t}$, current state $s_t$, dimension of action $k$, rejection rate $\delta > 0$
2: **Output:** The sampled action $a_t^{1:k}$
3: Let $x_t \leftarrow \mathtt{concat}(\tau_{<t}, s_t)$
4: **for** $i = 1, ..., k$ **do**
5:     Compute $p_{\mathrm{GPT}}(a_t^i \mid x_t, a_t^{<i})$       Note that $a_t^{<1} = \varnothing$.
6:     Compute $p_{\mathrm{TPM}}(V_t \mid x_t, a_t^{<i}) = \sum_{a_t^{i:k}} p_{\mathrm{TPM}}(V_t, a_t^{i:k} \mid x_t, a_t^{1:k})$    $\triangleright$ The marginal can be efficiently computed by PC in linear time. See the algorithm in Appx. B.1.
7:     Compute $v_\delta = \mathtt{max}_v\{v \in \mathtt{val}(V_t) \mid p_{\mathrm{TPM}}(V_t \geq v \mid x_t, a_t^{<i}) \geq 1 - \delta\}$, for each $a_t^i \in \mathtt{val}(A_t^i)$
8:     Compute $\tilde{p}(a_t^i \mid x_t, a_t^{<i}; v_\delta) = \frac{1}{Z} \cdot p_{\mathrm{GPT}}(a_t^i \mid x_t, a_t^{<i}) \cdot p_{\mathrm{TPM}}(V_t \geq v_\delta \mid x_t, a_t^{<i})$ $\triangleright$ Apply Equation (2)
9:     Sample $a_t^i \sim \tilde{p}(a_t^i \mid x_t, a_t^{<i}; v_\delta)$
10: **end for**
11: **return** $a_t^{1:k}$

---

---

**Algorithm 3** Multi-step Value Estimate

---

1: **Input:** $\tau_{\leq t} = (s_0, a_0, ..., s_t, a_t)$, sequence model $\mathcal{M}$, terminal timestep $t' > t$, discount $\gamma$
2: **Output:** The multi-step value estimate $\mathbb{E}[V_t^m]$
3: Sample future actions $a_{t+1}, ..., a_{t'}$ from $\mathcal{M}$
4: Compute $p_{\mathrm{TPM}}(r_h \mid \tau_{\leq t}, a_{t+1:h}) = \sum_{s_{t+1:h}} p_{\mathrm{TPM}}(r_h, s_{t+1:h} \mid \tau_{\leq t'})$ for $h \in [t+1, t']$    $\triangleright$ Marginalize over intermediate states $s_{t+1:h}$
5: Compute $p_{\mathrm{TPM}}(\mathrm{RTG}_{t'} \mid \tau_{\leq t}, a_{t+1:t'}) = \sum_{s_{t+1:t'}} p_{\mathrm{TPM}}(\mathrm{RTG}_{t'} \mid \tau_{\leq t'})$
6: Compute

$$\mathbb{E}[V_t^m] = \sum_{h=t}^{t'} \gamma^{h-t} \mathbb{E}_{r_h \sim p_{\mathrm{TPM}}(\cdot \mid \tau_{\leq t}, a_{t+1:h})}[r_h] + \gamma^{t'+1-t} \mathbb{E}_{\mathrm{RTG}_{t'} \sim p_{\mathrm{TPM}}(\cdot \mid \tau_{\leq t}, a_{t+1:t'})}[\mathrm{RTG}_{t'}]$$

7: **return** $\mathbb{E}[V_t^m]$

---

## C.2 Computing Multi-Step Value Estimates

In this section, we present an efficient algorithm that computes Equation (3). From the decomposition of Equation (3), we can calculate $\mathbb{E}[V_t^m]$ if we have the probabilities $p(r_\tau \mid s_t, a_{t:t'})(t \leq \tau < t')$ and $p(\mathrm{RTG}_{t'} \mid s_t, a_{t:t'})$. A simple approach would be to compute each of the $t' - t + 1$ probabilities separately using the algorithm described in Appx. B.1 (recall that conditional probabilities are quotient of the corresponding marginal probabilities). However, this approach has an undesired time complexity that scales linearly with respect to $t' - t + 1$.

Following [28], we describe an algorithm that can compute all desired quantities using a single feedforward and a backward pass to the PC.

**The forward pass.** The forward pass is similar to the one described in Appx. B.1. Specifically, we set the evidence as $s_t, a_{t:t'}$ and execute the forward pass.

**The backward pass.** The backward pass consists of two steps: (i) traverse all nodes parents before children to compute a statistic termed flow for every node $n$: $\mathtt{flow}_n$; (ii) compute the target probabilities using the flow of all input nodes. Recall that we assume without loss of generality that PCs alternate between sum and product layers. We further assume that all parents of input nodes are product nodes. We define the flow of the root node as 1. The flow of other nodes is defined recursively as (define $p_n$ as the forward probability of node $n$):

$$
\mathtt{flow}_n := \begin{cases} \sum_{m \in \mathtt{pa}(n)} (\theta_{m,n} \cdot p_n/p_m) \cdot \mathtt{flow}_m & n \text{ is a product node,} \\ \sum_{m \in \mathtt{pa}(n)} \mathtt{flow}_m & n \text{ is a input or sum node,} \end{cases}
$$

where $\mathtt{pa}(n)$ is the set of parents of node $n$.

Next for every variable $X \in \{R_t, \ldots, R_{t'-1}, \mathrm{RTG}_{t'}\}$, we first collect all input nodes defined on $X$. Define the set of input nodes as $S$. We have that

$$
p(x|s_t, a_{t:t'}) := \frac{1}{Z} \sum_{n \in S} \mathtt{flow}_n \cdot f_n(x),
$$

where $f_n$ is defined in Equation (7) and $Z$ is a normalizing constant.

## D  Inference-time Optimality Score

We define the inference-time optimality score as a proxy for inference-time optimality. This score is primarily defined over a state-action pair $(s_t, a_t)$ at each inference step. In Figure 2 (middle) and Figure 2 (right), each sample point represents a trajectory, and the corresponding inference-time optimality score is defined over the entire trajectory by averaging the scores of all inference steps.

The specific steps for calculating the score for a given inference step $t$, given $s_t$ and a policy $p(a_t \mid s_t)$, are as follows:

1. Given $s_t$, sample $a_t$ from $p_{\mathrm{TT}}(a_t \mid s_t)$, $p_{\mathrm{DT}}(a_t \mid s_t)$, or $p_{\mathrm{Trifle}}(a_t \mid s_t)$.
2. Compute the state-conditioned value distribution $p^s(V_t \mid s_t)$.
3. Compute $R_t := \mathbb{E}_{V_t \sim p^a(\mathrm{RTG}_t | s_t, a_t)}[V_t]$, which is the corresponding estimated expected value.
4. Output the quantile value $S_t$ of $R_t$ in $p^s(V_t \mid s_t)$.

To approximate the distributions $p^s(V_t \mid s_t)$ and $p^a(V_t \mid s_t, a_t)$ (where $V_t = \mathrm{RTG} * t$) in steps 2 and 3, we train two auxiliary GPT models using the offline dataset. For instance, to approximate $p^s(V_t \mid s_t)$, we train the model on sequences $(s * t - k, V_{t-k}, \ldots, s_t, V_t)$.

Intuitively, $p^s(V_t \mid s_t)$ approximates $p(V_t \mid s_t) := \sum_{a_t} p(V_t \mid s_t, a_t) \cdot p(a_t \mid s_t)$. Therefore, $S_t$ indicates the percentile of the sampled action in terms of achieving a high expected return, relative to the entire action space.

## E  Additional Experimental Details

### E.1  Gym-MuJoCo

**Sampling Details.** We take the single-step value estimate by setting $V_t = \mathrm{RTG}_t$ and sample $a_t$ from Equation (2). When training the GPT used for querying $p_{\mathrm{GPT}}(a_t^i | s_t, a_t^{<i})$, we adopt the same model specification and training pipeline as TT or DT. When computing $p_{\mathrm{TPM}}(V_t \geq v | s_t, a_t^{\leq i})$, we first use the learned PC to estimate $p(V_t | s_t)$ by marginalizing out intermediate actions $a_{t:t'}$ and select the $\epsilon$-quantile value of $p(V_t | s_t)$ as our prediction threshold $v$ for each inference step. Empirically we fixed $\epsilon$ for each environment and $\epsilon$ ranges from 0.1 to 0.3.

**Beam Search Hyperparameters.** The maximum beam width $N$ and planning horizon $H$ that Trifle uses across 9 MuJoCo tasks are 15 and 64, respectively.

**Comparison with Value-Based Algorithms.** To shed light on how Trifle compares to methods that directly optimize the Q values while filtering actions by conditioning on high returns (as done in RvS algorithms), we compare Trifle with Q-learning Decision Transformer [46], which incorporates a contrastive Q-learning regime into the RvS framework. As shown in the table below, Trifle outperforms QDT in all six adopted MuJoCo benchmarks:

Table 3: Normalized Scores of QDT and Trifle on Gym-MuJoCo benchmarks

| Dataset | Environment | Trifle | QDT |
|---|---|---|---|
| Medium | Halfcheetah | $\mathbf{49.5}_{\pm 0.2}$ | $42.3_{\pm 0.4}$ |
| Med-Replay | Halfcheetah | $\mathbf{45.0}_{\pm 0.3}$ | $35.6_{\pm 0.5}$ |
| Medium | Hopper | $\mathbf{67.1}_{\pm 4.3}$ | $66.5_{\pm 6.3}$ |
| Med-Replay | Hopper | $\mathbf{97.8}_{\pm 0.3}$ | $52.1_{\pm 20.3}$ |
| Medium | Walker2d | $\mathbf{83.1}_{\pm 0.8}$ | $67.1_{\pm 3.2}$ |
| Med-Replay | Walker2d | $\mathbf{88.3}_{\pm 3.8}$ | $58.2_{\pm 5.1}$ |

### E.2 Stochastic Taxi Environment

**Hyperparameters.** Except for s-Trifle, the sequence length $K$ modeled by TT, DT, and m-Trifle is all equal to 7. The inference algorithm of TT follows that of the MuJoCo experiment and DT follows its implementation in the Atati benchmark. Notably, during evaluation, we condition the pretrained DT on 6 different RTGs ranging from -100 to -350 and choose the best policy resulting from RTG=-300 to report in Figure 3c. Beam width $N = 8$ and planning horizon $H = 3$ hold for TT and m-Trifle.

**Additional Results on the Taxi benchmark.** Besides the episode return, we adopt two metrics to better evaluate the adopted methods: (i) #penalty: the average number of executing illegal actions within an episode; (ii) $P(\texttt{failure})$: the probability of failing to transport the passenger to the destination within 300 steps.

Table 4: Results on the stochastic Taxi environment. All the reported numbers are averaged over 1000 trials.

| Methods | Episode return | # penalty | $P(\text{failure})$ |
|---|---|---|---|
| s-Trifle | -99 | 0.14 | 0.11 |
| m-Trifle | **-57** | 0.38 | **0.02** |
| TT | -182 | 2.57 | 0.34 |
| DT | -388 | 14.2 | 0.66 |
| DoC | -146 | **0** | 0.28 |
| dataset | -128 | 2.41 | 0 |

**Ablation Study Regarding Action Filtering.** In an attempt to justify the effectiveness/necessity of exact inference, we compare Trifle with value-based action filtering/value estimation in the following:

To begin with, we implemented the traditional Policy Evaluation algorithm on the Taxi offline dataset described in Section 6.2 of the paper. The policy evaluation algorithm is based on the Bellman update:

$$Q(s_t, a_t) \leftarrow Q(s_t, a_t) + \alpha \big[ r_{t+1} + \gamma Q(s_{t+1}, a_{t+1}) - Q(s_t, a_t)) \big]$$

Then we use the obtained Q function, denoted $Q_{\text{taxi}}$, to perform the following ablation studies. We still choose TT as our base RvS model. Recall that given $s_t$, TT first samples $a_t$ from its learned prior policy $p_{TT}(a_t|s_t)$, which are subsequently fed to a beam search procedure that uses the learned value function $p_{TT}(V_t|s_t, a_t)$ to select the best action. Therefore, we consider ablations on two key components of TT: (i) the prior policy $p_{TT}(V_t|s_t, a_t)$ used to sample actions, and (ii) the value function $p_{TT}(V_t|s_t, a_t)$ used to evaluate and select actions.

1. **TT + $Q_{\textbf{taxi}}$ action filtering**: weigh the prior policy $p_{TT}(a_t|s_t)$ with each action's exponentiated $Q_{\text{taxi}}$ value (i.e., $exp(Q_{\text{taxi}}(s_t, a_t))$), but still adopt TT's value estimation. In other words, in this experiment, we only use $Q_{\text{taxi}}$ to improve the sampling quality as s-Trifle does.

2. **TT + $Q_{\textbf{taxi}}$ value estimation**: replace $p_{TT}(V_t|s_t, a_t)$ with $Q_{\text{taxi}}(s_t, a_t)$ for action evaluation and selection, but still use TT's prior policy $p_{TT}(a_t|s_t)$.

3. **TT + full $Q_{\textbf{taxi}}$**: simultaneously use $exp(Q_{\text{taxi}}(s_t, a_t))$ for action filtering and $Q_{\text{taxi}}(s_t, a_t)$ for action evaluation.

We present the results of these ablation studies as follows:

| Method | Score |
|---|---|
| TT | -182 |
| TT + $Q_{\text{taxi}}$ action filtering | -157 |
| TT + $Q_{\text{taxi}}$ value estimation | -147 |
| TT + full $Q_{\text{taxi}}$ | -138 |
| m-Trifle | -58 |
| s-Trifle | -99 |

From these results, we draw the following conclusions:

- m-Trifle and s-Trifle achieve the best performance.

- The rank of scores: TT + full $Q_{\text{taxi}}$ > TT + $Q_{\text{taxi}}$ value estimation > TT + $Q_{\text{taxi}}$ action filtering > TT suggests that using $Q_{\text{taxi}}$ for both action filtering and value estimation is beneficial; combining the two leads to the best performance.

- Specifically, the fact that s-Trifle outperforms the $Q_{\text{taxi}}$ based action filtering demonstrates that our filtration with exact inference is much more effective. The superior performance of m-Trifle also provides strong evidence that explicit marginalization over future states leads to better value estimation.

# F    Additional Experiments

## F.1    Ablation Studies on Rejection Sampling and Beam Search

The key insight of Trifle to solve challenges elaborated in Scenario #1 is to utilize tractable probabilistic models to better approximate action samples from the desired distribution $p(a_t|s_{0:t}, \mathbb{E}[V_t] \geq v)$. We highlight that the most crucial design choice of our method for this goal is that: Trifle can effectively bias the per-action-dimension generation process of any base policy towards high expected returns, which is achieved by adding per-dimension correction terms $p_{TPM}(V_t \geq v|s_t, a_t^{\leq i})$ (Eq. (2) in the paper) to the base policy.

While the rejection sampling method can help us obtain more unbiased action samples through a post value(expected return)-estimation session, we only implement this component for TT-based Trifle (not for DT-based Trifle) for fair comparison, as the DT baseline doesn't perform explicit value estimation or adopt any rejection sampling methods. Therefore, the success of DT-based Trifle strongly justifies the effectiveness of the TPM components. Moreover, the beam search algorithm also comes from TT. Although it is a more effective way to do rejection sampling, it is not the necessary component of Trifle, either.

Figure 5: Scaling Curves of Inference Time. (Fix beam width = 32)

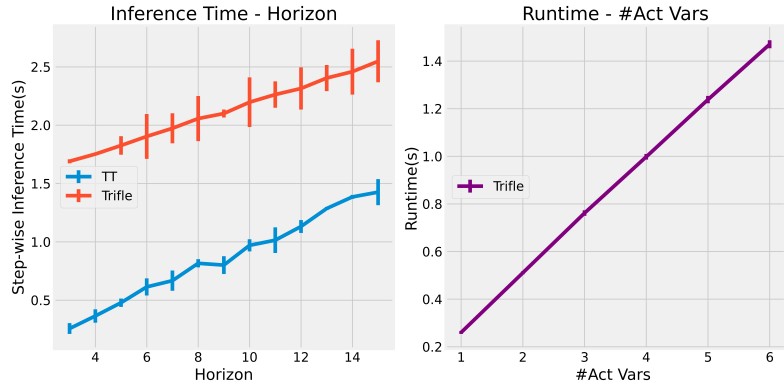

Table 5: Ablations over Beam Search Hyperparameters on Halfcheetah Med-Replay. (a) With $H = 1$, the beam search degrades to naive rejection sampling (b) With $W = 1$, the algorithm doesn't perform rejection sampling. It samples a single action and applies it to the environment directly.

<table>
<tr><td colspan="4" align="center">(a) Varying Planning Horizon</td><td colspan="4" align="center">(b) Varying Beam Width</td></tr>
<tr><td>Horizon $H$</td><td>Width $W$</td><td>TT</td><td>TT-based Trifle</td><td>Horizon $H$</td><td>Width $W$</td><td>TT</td><td>TT-based Trifle</td></tr>
<tr><td>5</td><td>32</td><td>41.9±2.5</td><td>45.0±0.3</td><td>5</td><td>32</td><td>41.9±2.5</td><td>45.0±0.3</td></tr>
<tr><td>4</td><td>32</td><td>40.1±2.0</td><td>43.1±1.0</td><td>5</td><td>16</td><td>42.5±1.9</td><td>42.6±1.6</td></tr>
<tr><td>3</td><td>32</td><td>41.6±1.3</td><td>42.6±1.6</td><td>5</td><td>8</td><td>42.9±0.4</td><td>43.5±0.3</td></tr>
<tr><td>2</td><td>32</td><td>39.7±2.5</td><td>42.8±0.5</td><td>5</td><td>4</td><td>38.7±0.3</td><td>43.4±0.3</td></tr>
<tr><td>1 (w/ naive rej sampling)</td><td>32</td><td>33.6±3.0</td><td>39.6±0.7</td><td>5</td><td>1 (w/o rej sampling)</td><td>31.2±3.4</td><td>36.7±1.8</td></tr>
</table>

For TT-based Trifle, we adopted the same beam search hyperparameters as reported in the TT paper. We conduct ablation studies on beam search hyperparameters in Table 5 to investigate the effectiveness of Trifle's each component. From Table 5, we can observe that:

- Trifle consistently outperforms TT across all beam search hyperparameters and is more robust to variations of both planning horizon and beam width.

- (a) Trifle w/ naive rejection sampling » TT w/ naive rejection sampling (b) Trifle w/o rejection sampling » TT w/o rejection sampling. In both cases, Trifle can positively guide action generation.

- Trifle w/ beam search > Trifle w/ naive rejection sampling > Trifle w/o rejection sampling » TT w/ naive rejection sampling. Although other design choices like rejection sampling/beam search help to better approximate samples from the high-expected-return-conditioned action distribution, the per-dimension correction terms computed by Trifle play a very significant role.

### F.2 Computational Efficiency Analysis

Since TPM-related computation consistently requires 1.45s computation time across different horizons, the relative slowdown of Trifle is diminishing as we increase the beam horizon. Specifically, in the Gym-Mujuco benchmark, the time consumption for one step (i.e., one interaction with the environment) of TT and Trifle with different beam horizons are listed here (Figure 5 (left) also plots an inference-time scaling curve of Trifle vs TT with varying horizons):

Table 6: The one-step inference runtime of the Gym-MuJuCo benchmark

| Horizon | TT | Trifle |
|---------|------|--------|
| 5 | 0.5s | 1.5s |
| 15 | 1.2s | 1.8s |

Moreover, Figure 5 (right) shows that Trifle's runtime (TPM-related) scales *linearly* w.r.t. the number of action variables, which indicates its efficiency for handling high-dimensional action spaces.

Trifle is also efficient in training. It only takes 30-60 minutes ( 20s per epoch, 100-200 epochs) to train a PC on one GPU for each Gym-Mujuco task (*Note that a single PC model can be used to answer all conditional queries required by Trifle*). In comparison, training the GPT model for TT takes approximately 6-12 hours (80 epochs).

### F.3 Ablation Studies on the Adaptive Thresholding Mechanism

The adaptive thresholding mechanism is adopted when computing the term $p_{TPM}(V_t \geq v | s_t, a_t^{\leq i})$ of Equation (2), where $i \in \{1, \ldots, k\}$, $k$ is the number of action variables and $a_t^i$ is the $i$th variable of $a_t$. Instead of using a fixed threshold $v$, we choose $v$ to be the $\epsilon$-quantile value of the distribution $p_{TPM}(V_t | s_t, a_t^{<i})$ computed by the TPM, which leverage the TPM's ability to exactly compute marginals given **incomplete** actions (marginalizing out $a_t^{i:k}$). Specifically, we compute $v$ using $v = max_r\{r \in \mathbb{R} | p_{TPM}(V_t \geq r | s_t, a_t^{<i}) \geq 1 - \epsilon\}$. Empirically we fixed $\epsilon$ for each Gym-MuJoCo environment and $\epsilon = 0.2$ or $0.25$, which is selected by running grid search on $\epsilon \in [0.1, 0.25]$.

Table 7: Comparison of Adaptive and Fixed Thresholding Mechanisms

(a) Ablations over Adaptive Thresholding (Varying $\epsilon$) on Halfcheetah Med-Replay

| Method | Score |
|---|---|
| TT | 41.9±2.5 |
| TT-based Trifle ($\epsilon = 0.25$) | **45.0±0.3** |
| TT-based Trifle ($\epsilon = 0.2$) | 44.2±0.4 |
| TT-based Trifle ($\epsilon = 0.15$) | 44.4±0.3 |
| TT-based Trifle ($\epsilon = 0.1$) | 42.6±1.6 |

(b) Performance of Fixed Thresholding (Varying $v$)

| $v$ | Halfcheetah Med-Replay | Walker2d Med-Expert |
|---|---|---|
| adaptive | **45.0±0.3** | **109.3±0.1** |
| 90 | 44.8±0.3 | 109.1±0.2 |
| 80 | 39.5±2.8 | 108.9±0.2 |
| 70 | 44.9±0.3 | 108.4±0.4 |
| 60 | 42.6±1.6 | 105.8±0.3 |
| 50 | 41.4±2.0 | 107.5±1.5 |
| 40 | 42.6±1.6 | 107.5±1.4 |
| 30 | 44.0±0.4 | 98.3±5.4 |

We report the performance of TT-based Trifle with variant $\epsilon$ vs TT on Halfcheetah Med-Replay benchmark in Table 7a. We can see that Trifle is robust to the hyperparameter $\epsilon$ and consistently outperforms the base policy TT.

We also conduct ablation studies comparing the performance of the adaptive thresholding mechanism with the fixed thresholding mechanism on two environments in Table 7b. Specifically, given that $V_t$ is discretized to be a categorical variable with 100 categoricals (0-99), we fix $v$ to be 90,80,70,60,50,40,30 respectively.

The table shows that the adaptive approach consistently outperforms the fixed value threshold in both environments. Additionally, the performance variation of fixing $v$ is larger compared to fixing $\epsilon$ as different $v$ can be optimal for different states.

## G Potential Negative Societal Impact

This paper proposes a new offline RL algorithm, which aims to produce policies that achieve high expected returns given a pre-collected dataset of trajectories generated by some unknown policies. When there are malicious trajectories in the dataset, our method could potentially learn to mimic such behavior. Therefore, we should only train the proposed agent in verified and trusted offline datasets.

