# OpenReview forum: "A Tractable Inference Perspective of Offline RL"
_NeurIPS.cc/2024/Conference — NeurIPS 2024 poster_

### Official Review · Reviewer_TgpP · 2024-07-07

**Soundness:** 3
**Presentation:** 3
**Contribution:** 2
**Rating:** 6
**Confidence:** 3

**Summary:**

This paper considers the RvS setting, where a sequence model is learned and used for a control-as-inference policy extraction. Naive sequence models fail due to overly optimistic reward-to-go in stochastic environments. It is possible to alleviate this with rejection sampling, but this suffers from the curse of dimensionality. This paper proposes the use of Tractable Probabilistic Models, which allow the evaluation of probabilistic queries given various degrees of observability. The proposed method uses a beam search procedure to evaluate sequences of actions to locate the best sequence. Results are presented on D4RL Mujoco benchmarks and stochastic game environments.

**Strengths:**

- The paper identifies a key issue in RvS methods, and proposes a new model to address it. It takes advantages of TPM models to evaluate returns under various degrees of masking.
- The paper is written in a clear way.
- Empirical results are promising, showing a consistent improvement in both deterministic and stochastic benchmarks.
- The method is agnostic to the base policy distribution used, meaning it can be easily adapted.

**Weaknesses:**

- It seems like the strength of TPM comes from beam search conducted on trajectories sampled from a separate policy model (e.g. TT, DT). This comes with the downside of a more costly inference and requires training multiple models. Is there a baseline comparison that utilizes the same computational efficiency?
- Is there a simple baseline to implement for the beam search value estimator that does not require a specialized TPM? To be thorough, it would be nice to decouple the beam search idea from the use of a TPM.
- It would be good to define RvS abbreviation early on.
- More detail on the formulation of the TPM model would clarify the method. How is the model trained and with what architecture?

**Questions:**

See weaknesses above.

**Limitations:**

A brief limitations section is present, which somewhat addresses the main limitations of the method.

---

> ### Author Rebuttal · Authors · 2024-08-07
>
> ### Comment #1: whether beam search is important in Trifle
>
> we will first clarify the relationship between Trifle and beam search. Then we conduct a comprehensive ablation study on beam search algorithms to confirm Trifle's superiority.
>
> As mentioned in the general response, the key insight of Trifle to solve challenge #1 is to utilize tractable probabilistic models to better approximate action samples from the desired distribution $p(a_t | s_{0:t}, \mathbb{E}[V_t] \geq v)$. We highlight that the most crucial design choice of our method for this goal is that: Trifle can effectively bias the per-action-dimension generation process of any base policy towards high expected returns, which is achieved by adding per-dimension correction terms $p_{TPM} (V_t \geq v | s_t, a_{t}^{\leq i})$ (Eq. (2) in the paper) to the base policy.
>
> While the rejection sampling method can help us obtain more unbiased action samples through a post value(expected return)-estimation session, we only implement this component for TT-based Trifle (not for DT-based Trifle) for fair comparison, as the DT baseline doesn't perform explicit value estimation or adopt any rejection sampling methods. Therefore, the success of DT-based Trifle strongly justifies that the correction term computed by TPM can effectively bias the actions generated by DT towards higher expected returns.
>
> Moreover, the beam search algorithm comes from TT.  Although it is a more effective way to do rejection sampling, it is not the necessary component of Trifle.
>
> **Ablations over beam search/rejection sampling**
>
> For TT-based Trifle, we adopted the same beam search hyperparameters as reported in the TT paper (in the official GitHub repo https://github.com/jannerm/trajectory-transformer). And we conduct ablation studies on beam search hyperparameters in **Table 2** of PDF to investigate the effectiveness of Trifle's each component:
>
> - Trifle consistently outperforms TT across all beam search hyperparameters and is more robust to variations of both planning horizon $H$ and beam width $W$.
> - (a)Trifle w/ naive rejection sampling >> TT w/ naive rejection sampling (b) Trifle w/o rejection sampling >> TT w/o rejection sampling. In both cases, Trifle's superior performance originates from that it can positively guide action generation (similar to DT-based Trifle vs DT).
> - Trifle w/ beam search > Trifle w/ naive rejection sampling > Trifle w/o rejection sampling >> TT w/ naive rejection sampling. Although other design choices like rejection sampling/beam search help to better approximate samples from the desired distribution $p(a_t | s_{0:t}, \mathbb{E}[V_{t}] \geq v)$, the per-dimension correction terms computed by Trifle used to guide per-dimension-action generation plays a very significant role.
>
> ### Comment #2: the downside of a more costly inference and requires training multiple models.
>
> Trifle is efficient in training. It only takes 30-60 minutes (~20s per epoch, 100-200 epochs) to train a PC on one GPU for each Gym-Mujuco task (**Note that a single PC can be used to answer all conditional queries required by Trifle**). In comparison, training the GPT model for TT takes approximately 6-12 hours (80 epochs).
>
> To evaluate inference-time efficiency, we conduct a more detailed runtime analysis, and the main results are shown in Figure 1 of the attached PDF. Figure 1 (left) expands Table 5 and plots the stepwise inference-time scaling curve of Trifle vs TT with varying horizons. We can see that: As we increase the horizon, the relative slowdown is mitigated. This is because, across different horizons, **TPM-related computation consistently requires ~1.45s computation time**, which additional computation overhead is diminishing as we increase the beam horizon. Figure 1 (right) shows that Trifle's runtime (TPM-related) scales **linearly** w.r.t. the number of action variables, which indicates its efficiency for handling high-dimensional action spaces.
>
> Note that there are recent breakthroughs [1] in designing efficient PC implementations, which can significantly speed up the computation of Trifle (both training and inference).
>
> [1] Liu, Anji, Kareem Ahmed, and Guy Van den Broeck. "Scaling Tractable Probabilistic Circuits: A Systems Perspective." arXiv preprint arXiv:2406.00766 (2024).
>
> ### Comment #3: define the RvS abbreviation earlier
>
> We thank the reviewer for the suggestion and will revise the paper accordingly.
>
> ### Comment #4: details on the formulation, structure, and training algorithm of the TPM model
>
> We thank the reviewer for their suggestion. We provide a concise and intuitive introduction to the TPM we adopted: Probabilistic Circuits (PCs). We describe the representation and structure of PCs in Section 2 and provide a detailed and formal introduction in Appendix B. We adopted the Hidden Chow-Liu Tree PC structure and used EM to train the model. Please refer to Appendix B.2 for more details about the adopted PC structures and the parameter learning algorithm.

---

> > ### Comment · Reviewer_TgpP · 2024-08-12
> >
> > Thank you to the authors for a detailed response. These clarifications are helpful in giving a more clear picture of the method, and I would encourage the authors to add these rationales into the experimental setup section. The description of the TPM model in Appendix B is brief, and the paper would benefit from a more thorough description as the average RL practitioner would be unaware of the details of the TPM model. I think this is a solid paper integrating the probabilistic circuits framework into RL, and I will opt to maintain my current score.

---

> ### Author Response · Authors · 2024-08-13
>
> Thank you for your thoughtful feedback and appreciation of our response, and we appreciate your recognition of our work as solid. We will revise the paper to include the additional experimental details and a more comprehensive description of TPMs, as suggested. If you have any further questions, we are pleased to discuss them with you.

---

### Official Review · Reviewer_dmuC · 2024-07-08

**Soundness:** 3
**Presentation:** 2
**Contribution:** 2
**Rating:** 6
**Confidence:** 3

**Summary:**

The paper provides empirical evidence that sequence models are able to identify promising actions, but that their policies at inference-time can be suboptimal. The paper proposes to use a tractable probabilistic model to bias the generated action sequences towards optimal ones. The paper provides fairly extensive empirical validation for its claims.

I think the paper could benefit from merging and rephrasing Sections 4 and 5. In its current state, I found it difficult to gather the proposed algorithm. I would also consider adding a (reduced) algorithm box to the main paper.

**Strengths:**

* The experimental evaluation of the claims is thorough and includes a wide selection of relevant baselines.
* Trifle appears to be on par or slightly better than a number of other approaches.

**Weaknesses:**

* It is mentioned that a viable alternative to Trifle are sequence models augmented by components learning the Q-function (e.g., QDT). However, Trifle is not compared empirically against these methods. I believe such a comparison would significantly strengthen the paper.
* I find Figure 2 (middle) slightly misleading. Why does it not directly plot the actual returns (and instead the "optimality score")? The correlation between "optimality score" and actual returns appears to be pretty weak in the relevant range. Why are middle and right not merged into one single figure?
* In the Gym-MuJoCo benchmarks, DD (conditional generative modeling) seems to perform roughly as well as Trifle. I believe that a more extensive comparison between the approaches would benefit the paper. Why would Trifle be preferable?

**Questions:**

* In Figure 2, what is the definition of the "optimality score"?
* Why is the formulation of eq. (1) preferred over training a policy $\pi$ which maximizes expected return plus a (KL-)regularization term that encourages it to stay close to the data-generating distribution (such as common in the RLHF literature, e.g., [1])?
* How do you choose $t'$?
* Why is the additional rejection sampling step (lines 206-207) necessary given that $\tilde{p}$ is already proportional to the probability of having large multi-step value?
* How does the efficiency compare against conditional generative modeling (e.g., DD) and approaches such as QDT, given that TPMs can be slow to train?


[1]: Rafailov, Rafael, et al. "Direct preference optimization: Your language model is secretly a reward model." Advances in Neural Information Processing Systems 36 (2024).

**Limitations:**

See weaknesses and questions above.

---

> ### Author Rebuttal · Authors · 2024-08-07
>
> ### Comment #1: comparison with models augmented by Q-function learning components (e.g., QDT)
>
> We compare the TT-based Trifle with QDT in Appendix 4.1, and DT-based Trifle with QDT in **Table 3(a)** of the rebuttal PDF. The results demonstrate that DT-based Trifle significantly outperforms QDT, supporting our claim that in such scenarios, improving inference-time performance is more critical.
>
> Additionally, since QDT enhances the training side with more accurate training labels and Trifle improves the inference side with better action sampling, both methods could be combined to achieve even better performance.
>
> ### Comment #2: definition of the inference-time optimality score
>
> We define the "inference-time optimality score" as a proxy for inference-time optimality. This score is primarily defined over a state-action pair $(s_t, a_t)$ at each inference step. In Figure 1 (middle) and Figure 1 (right), each sample point represents a trajectory, and the corresponding "inference-time optimality score" is defined over the entire trajectory by averaging the scores of all inference steps.
>
> The inference-time optimality score at time $t$ is defined by the quantile value of $R_t := \mathbb{E}_{V_t \sim p^a(\mathrm{RTG}_t \mid s_t, a_t)} [V_t]$ ($a_t$ is the action selected by the agent) under the distribution $p(V_t \mid s_t)$. This value will be high if the sampled action has a high estimated expected return. Please refer to the official comments for more details.
>
> ### Comment #3: the correlation between the optimality score and the actual returns
>
> (Following the above response)
> The above definition uses $p(V_t \mid s_t)$ ($V_t = RTG_t$) to measure the quality of $a_t$. The validity of this proxy is based on two assumptions. First, in environments like gym-mujoco, the $ RTG_t $ labels provided in the offline dataset are of high quality and are a strong indicator of actual return (justified by Figure 1 (left)). Second, the $p(V_t \mid s_t)$ fitted using GPT can accurately approximate $p(V_t \mid s_t) := \sum_{a_t} p(V_t \mid s_t, a_t) \cdot p(a_t \mid s_t)$. Given the high-dimensional action space, the second assumption is challenging to verify directly. However, Figure 1 (right) shows a clear positive correlation between this proxy and the actual return, indicating that trajectories with higher scores often achieve higher final returns, with the small slope largely attributable to scale differences.
>
> In summary, we have theoretically and empirically verified the effectiveness of the "inference-time optimality score" as a proxy. Figure 1 (middle) compares the inference-time performance of different policies by visualizing the distribution of the "inference-time optimality score" over multiple trajectories.
>
> ### Comment #4: comparison with DD
>
> We thank the reviewer for drawing our attention to the interesting connection between Trifle and DD. First, as shown in **Table 3(b)** of the PDF, we highlight that Trifle outperforms DD on 7 out of 9 MuJoCo benchmarks in Table 1 and is more robust with smaller stds.
>
> On the methodology side, DD adopts diffusion models to fit state-action sequences and then generates rewarding sequences by conditioning on high return. From the inference side, similar to DT, DD aims to sample action conditions on high return, while Trifle aims to improve upon existing RvS algorithms (e.g., TT and DT) by allowing them to sample action conditions on the *expected* return. While DD achieves superior performance compared to some other baselines, their main technical contribution is orthogonal to that of Trifle.
>
> Next, in Section 4.2, we highlight Trifle's effectiveness in handling stochastic environments by efficiently and exactly computing the multi-step value estimate (Eq. (3)). This can significantly improve value estimation in stochastic environments. Since DD also requires a $p (V | s_{0:t}, a_{0:t})$ to guide action sampling, this value component of Trifle can be potentially incorporated into DD. We will include the above discussions in the next version of the paper.
>
> ### Comment #5: connection with methods that train an entropy-regularized policy such as DPO
>
> We thank the reviewer for their insightful comment.
>
> In our humble opinion, we do not need to prefer one over the other as a policy trained with better objectives (e.g., that trained by some value-based offline RL methods) can be further utilized by RvS algorithms to obtain even better policies. For example, by using the Q values from a pre-trained IQL agent (which implies a better policy) for value estimation, TT can be significantly improved to be much better than both the vanilla TT and IQL.
>
> In the case of Trifle, the question is whether we want to pay the additional inference-time overhead to achieve better performance. According to the general response, the additional components of Trifle take about 1.45s, which is negligible compared to the runtime of the base algorithm.
>
> ### Comment #6: how to choose t'
>
> We choose $t' = t+3$. See the comments below for a detailed elaboration.
>
> ### Comment #7: why is the additional rejection sampling step necessary in lines 206-207
>
> The necessity of rejection sampling comes from the hardness to draw exact action samples conditioned on high expected returns, which is shown in Theorem 1. For m-Trifle in stochastic envs, we need to compute $\mathbb{E}[ V_t^{\mathrm{m}} \big ]$ using TPM via a post-value estimation step and perform rejection sampling to obtain more unbiased samples from the desired distribution in Eq. (1).
>
> ### Comment #8: training time of Trifle and baseline methods
>
> It takes 30-60 mins to train a PC (the adopted TPM) on one GPU for each Gym-Mujuco task. In comparison, training the GPT model for TT takes approximately 6-12 hours.
>
> ### Comment #9: detailed elaboration of the proposed algorithm
> Please refer to the official comments for a detailed response to this comment, thanks very much!

---

> > ### Comment · Reviewer_dmuC · 2024-08-07
> >
> > I thank the authors for their detailed comments to all reviewers and conducting a number of additional experiments and ablation studies.
> > I believe that these strengthen the paper and I increase my score to 6 accordingly.
> >
> > > Comment #3: the correlation between the optimality score and the actual returns
> >
> > Could the authors clarify why in Figure 2 (middle) they do not compare against the actual returns instead of the optimality proxy? In my understanding, the use of the proxy skews the results and makes them harder to interpret correctly. In particular, it appears to me that this biases the results in this figure to look more favorably for Trifle.

---

> > > ### Author Response · Authors · 2024-08-08
> > > **Further Clarifications about Figure 2**
> > >
> > > Thanks for the active feedback! Detailed comparisons against actual returns can be found in Table 1 of the paper, with Section 6.1 providing an in-depth analysis. Additionally, we have included ablation studies to validate the effectiveness of each component of Trifle in achieving superior performance.
> > >
> > > More importantly, the final performance is determined by both training-time and inference-time optimality. In Figure 2, our primary objective was to conceptually isolate the inference-time optimality issue and examine the inference-time performance of each method, thus highlighting our argument that existing RvS approaches underperform as a result of suboptimal inference-time performance.
> > >
> > > We hope this clarifies our intention. Thank you again for your insightful question.

---

> ### Author Response · Authors · 2024-08-07
> **Details of Inference-time Optimality Score**
>
> The specific steps for calculating the inference-time optimality score for a given inference step $t$, given $s_t$ and a policy $p(a_t \mid s_t)$, are as follows:
>
> 1. Given $s_t$, sample $a_t$ from $p_{TT}(a_t \mid s_t)$, $p_{DT}(a_t \mid s_t)$, or $p_{Trifle}(a_t \mid s_t)$.
> 2. Compute the state-conditioned value distribution $p^s(V_t \mid s_t)$.
> 3. Compute $R_t := \mathbb{E}_{V_t \sim p^a(RTG_t \mid s_t, a_t)} [V_t]$, which is the corresponding estimated expected value.
> 4. Output the quantile value $S_t$ of $R_t$ in $p^s(V_t \mid s_t)$.
>
> To approximate the distributions $p^s(V_t \mid s_t)$ and $p^a(V_t \mid s_t, a_t)$ (where $V_t = RTG_t$) in steps 2 and 3, we train two auxiliary GPT models using the offline dataset. For instance, to approximate $p^s(V_t \mid s_t)$, we train the model on sequences $(s_{t-k}, V_{t-k}, \ldots, s_t, V_t)$.
>
> Intuitively, $p^s(V_t \mid s_t)$ approximates $p(V_t \mid s_t) := \sum_{a_t} p(V_t \mid s_t, a_t) \cdot p(a_t \mid s_t)$. Therefore, $S_t$ indicates the percentile of the sampled action in terms of achieving a high expected return, relative to the entire action space.

---

> ### Author Response · Authors · 2024-08-07
> **Detailed Elaboration of the Proposed Algorithm**
>
> Thank the reviewer for the suggestion. We will include a subsection to describe the algorithm and also provide an algorithm table in the next version of the paper.
>
> The high-level idea of Trifle is that it can be built upon many RvS algorithms (e.g., TT, DT) to solve two key problems that are widely observed in the literature: (i) sample actions condition on high *expected* return, and (ii) estimate state-action values under stochastic transition dynamics. In the following, we first explain what modifications Trifle applied to solve both challenges; we then describe how the modifications are used to design algorithms upon existing RvS algorithms.
>
> **Scenario 1: sampling actions condition on high *expected* return**
>
> As described in Section 3, this problem can be formulated as: given a joint distribution over $p(s_{0:t}, a_{t}, V_{t})$, we want to query $p(a_t | s_{0:t}, \mathbb{E}[V_{t}] \geq v)$ (formally defined in Equation (1)).  However, Theorem 1 illustrates that it is NP-hard to exactly compute $p(a_t | s_{0:t}, \mathbb{E}[V_{t}] \geq v)$ even when $p(s_{0:t}, a_{t}, V_{t})$ follows a simple Naive Bayes distribution, which guides Trifle to find *good approximations* of the query.
>
> In general, we first train a PC (the adopted TPM) to fit the joint distribution over $p(s_{t-k}, a_{t-k}, V_{t-k},...,s_t, a_{t}, V_{t})$ from the offline dataset. This step is agnostic to the base RvS algorithm Trifle is built on. Then, given any component that computes or samples from $p(a_t | s_t, V_t = v)$ (DT) or $p(a_t | s_t)$ (TT), we replace it to a good approximation of $p(a_t | s_t, \mathbb{E}[V_t] \geq v)$ by augmenting these proposal distributions with a correction term.
>
> Specifically, we utilize the pretraind PC to compute the per-dimension correction term  $p_{TPM} (V_t \geq v | s_t, a_{t}^{\leq i})$ by marginalizing out unseen action variables $a_t^{i+1},...,a_t^k$ and this forms an exponentially large action space, aiming to bias the actions generated by the base policy towards high expected return $\mathbb{E}[V_t]$. Notably, we query this conditional probability for each dimension $a_t^i$, as the inference proceeds, the variable $i$ will increase progressively, resulting in different marginal queries. However, thanks to PCs' tractabiltiy, we can use a single model to answer arbitrary marginalization queries. Moreover, the computation can be done exactly without any approximation and scales **linearly** to the number of action variables, which is highly efficient. In Appendix C.2, we present the concrete algorithm of how PCs (the adopted TPM) compute conditional probabilities.
>
> Note that the rejection sampling methods via beam search steps is only adopted by the specific TT-based Trifle algorithm described in Section 4. DT doesn't apply beam search as TT so we don't implement rejection sampling for DT-based Trifle to ensure fair comparision.
>
> **Scenario 2: estimating state-action values under stochastic transition dynamics**
>
> Many existing RvS algorithms requires explicitly evaluating $p(V_t | s_{0:t}, a_{t})$. However, the single-step value estimate $V_t = \mathrm{RTG}_t$ will be very inaccurate under stochastic environments. This is because RTGs heavily depend on the randomness that determines state transitions. To combat this inaccuracy, it is common to use multi-step value estimates instead. TD(1) and TD(λ) are notable examples. TD(1) relies on full returns to update value estimates, while TD(λ) introduces a balance between n-step returns and full returns using a parameter λ, allowing for more accurate value estimation.
>
> While theoretically promising, classical multi-step value estimates generally require careful Monte Carlo sampling of different future state sequences, which makes it hard to obtain low-variance estimates. However, as introduced in Appendix C.2, with TPMs' ability to compute arbitrary conditional distributions, we can efficiently (in one forward pass and one backward pass of the TPM ) compute the multi-step value estimate in Eq. (3). This allows us to enjoy the better accuracy of the multi-step value estimates while mitigating the inefficiency to compute them.

---

> ### Author Response · Authors · 2024-08-07
> **Details of the m-Trifle algorithm and how we choose t'**
>
> For m-Trifle in stochastic Taxi environment, we largely follow the Algorithm 3 of Appendix 3.1, here's a more concise and complete version:
>
> (1) choose t'>t and compute $p_{GPT}(a_t | s_t)$, $p_{TPM}(V_{t'}>v | s_t, a_t)$
>
> (2) sample N action candidates from $p_{Trifle}(a_t|s_t) = p_{GPT}(a_t | s_t)\times p_{TPM}(V_{t'}>v | s_t, a_t)$
>
> (3) for each action candidate $a_t$, sample future actions $a_{t+1},...,a_t'$ autoregressively from $p_{Trifle}$ for future evaluation.
>
> (4) for each action candidate and each $h \in [t+1,t']$ , compute $p_{TPM} (r_h\ | s_t, a_{t:h})$  by marginalizing over intermediate states $s_{t+1:h}$ and also $p_{TPM} (V_{t'}\ | s_t, a_{t:t})$ , where $V_{t'} = RTG_{t'}$
>
> (5)  for each action candidate, compute：
> $E[ V_t^{m} ] = \sum_{h=t}^{t'} \gamma^{h-t} E_{r_h \sim p_{TPM} (\cdot | \tau_{\leq t}, a_{t+1:h})}  [ r_h ] + \gamma^{t'+1-t} E_{RTG_{t'} \sim p_{TPM} (\cdot | \tau_{\leq t}, a_{t+1:t'})}  [ V_{t'}  ]$
>
> (6) select the action with maximum $\mathbb{E}[ V_t^m ]$ to excute in the environment.
>
> We choose $t' = t+3$.

---

### Official Review · Reviewer_DwFR · 2024-07-13

**Soundness:** 2
**Presentation:** 3
**Contribution:** 2
**Rating:** 4
**Confidence:** 2

**Summary:**

This paper studies offline reinforcement learning and argues that tractability, e.g., the ability to answer probabilistic queries, is important for performance improvement. As a result, the authors propose a model that utilizes modern tractable generative models to answer arbitrary marginal/conditional probabilities. Comprehensive experiments demonstrate that the new model can achieve better performance in most cases.

**Strengths:**

The paper is overall well-written and easy to follow. The considered problem is interesting and relevant. A comprehensive set of experiments also verify the merits of the proposed approach.

**Weaknesses:**

While the proposed approach seems sound, the novelty and significance might be limited. The key idea in Sec 4 seems to be based on classical rejection sampling methods, though with some additions of correction terms.

Could the authors provide some discussions regarding computation complexity of the proposed methods?

**Questions:**

See above

**Limitations:**

Yes

---

> ### Author Rebuttal · Authors · 2024-08-07
>
> ### Comment #1: connection between rejection sampling and the correction terms, and the corresponding computational complexity
>
> We thank the reviewer for their constructive feedback. As mentioned in the general response, Trifle can be applied to many existing RvS algorithms (e.g., TT, DT) to mitigate the identified two challenges. To highlight, the per-action-dimension correction terms in Eq. (2) computed by TPM are crucial to enhance action sampling and mitigate the first challenge. And the rejection sampling methods via beam search steps, are only adopted by the specific TT-based Trifle algorithm described in Section 4. DT doesn't apply beam search as TT so we don't implement rejection sampling for DT-based Trifle. Therefore, the success of DT-based Trifle strongly justifies that the correction term can effectively bias the actions generated by DT towards higher expected returns.
>
>
> In the following, we will first discuss **how Trifle addresses both challenges** and why TPM matters, as well as the **computation complexity of the proposed methods**. Then we conduct ablation studies on rejection sampling to confirm Trifle's superiority.
>
> **Challenge 1: Sampling actions conditioned on high expected return**
>
> As described in Section 3, this problem can be formulated as: given a joint distribution over $p(s_{0:t}, a_{t}, V_{t})$, we want to query $p(a_t | s_{0:t}, \mathbb{E}[V_{t}] \geq v)$ (formally defined in Equation (1)). Note that the expectation over $V_t$ is very important because otherwise if we ignore the expectation operator and directly sample from $p(a_t | s_{0:t}, V_{t} \geq v)$, we may sample actions with a low probability that lead to high returns. This causes significant performance degradation as discussed in [1]. Then we naturally want to know whether we can exactly compute this quantity. However, Theorem 1 illustrates that it is NP-hard to exactly compute $p(a_t | s_{0:t}, \mathbb{E}[V_{t}] \geq v)$ even when $p(s_{0:t}, a_{t}, V_{t})$ follows a simple Naive Bayes distribution, which guides Trifle to find *good approximations* of the query.
>
> To achieve better approximations, we utilize TPM to compute per-action-dimension correction terms  $p_{TPM} (V_t \geq v | s_t, a_{t}^{\leq i})$ to bias the actions generated by any prior policy towards high expected return $\mathbb{E}[V_t]$. Note that we compute the correction term exactly for each dimension $a_t^i$, which is highly non-trivial as we have to marginalize out unseen action variables $a_t^{i+1},...,a_t^k$ and this forms an exponentially large action space. This is why intractable models suffer and we need to leverage the ability of TPMs to compute arbitrary conditional probabilities. Notably, the computation for TPM is exact without any approximation and scales **linearly** to the number of action variables, which is highly efficient. We will later show ablation results that confirm **the superiority of the correction term both with and without the rejection sampling step**.
>
> **Challenge 2: Estimating state-action values under stochastic transition dynamics**
>
> The second major challenge is the inaccuracy of the RTG labels when the environment transition dynamics are highly stochastic. This is because RTGs heavily depend on the randomness that determines state transitions. To combat this inaccuracy, it is common to use multi-step value estimates instead. TD(1) and TD(λ) are notable examples. TD(1) relies on full returns to update value estimates, while TD(λ) introduces a balance between n-step returns and full returns using a parameter λ, allowing for more accurate value estimation. An application of this is the Proximal Policy Optimization (PPO) algorithm, which uses the eligibility trace governed by TD(λ) to establish better value estimates. This approach helps PPO achieve more stable and accurate value estimation by effectively integrating future reward information over multiple timesteps.
>
> While theoretically promising, classical multi-step value estimates generally require careful Monte Carlo sampling of different future state sequences, which makes it hard to obtain low-variance estimates. However, as introduced in Appendix C.2, with TPMs' ability to compute arbitrary conditional distributions, we can efficiently (in one forward pass and one backward pass of the TPM ) compute the multi-step value estimate in Equation (3). This allows us to enjoy the better accuracy of the multi-step value estimates while mitigating the inefficiency of computing them.
>
> **Ablation studies on the rejection sampling/beam search**
>
> For TT-based Trifle, we adopted the same beam search hyperparameters as reported in the TT paper (in the official GitHub repo https://github.com/jannerm/trajectory-transformer). We conduct ablation studies on beam search hyperparameters in **Table 2** of PDF to investigate the effectiveness of Trifle's each component:
>
> -  Trifle consistently outperforms TT across all beam search hyperparameters and is more robust to variations of both planning horizon $H$ and beam width $W$.
> - (a)Trifle w/ naive rejection sampling >> TT w/ naive rejection sampling
>   (b) Trifle w/o rejection sampling >> TT w/o rejection sampling.
>   In both cases,  Trifle's superior performance originates from that it can positively guide action generation (similar to DT-based Trifle vs DT).
> - (a) Trifle w/ beam search > Trifle w/ naive rejection sampling > Trifle w/o rejection sampling >> TT w/ naive rejection sampling.
>   (b) Trifle w/ naive rejection sampling $\approx$ TT with beam search.
> Although other design choices like rejection sampling/beam search help to better approximate samples from the desired distribution $p(a_t | s_{0:t}, \mathbb{E}[V_{t}] \geq v)$, the per-dimension correction terms computed by Trifle used to guide per-dimension-action generation plays a very significant role.
>
> [1] Paster, Keiran, Sheila McIlraith, and Jimmy Ba. "You can’t count on luck: Why decision transformers and rvs fail in stochastic environments."

---

> ### Author Response · Authors · 2024-08-12
>
> Dear Reviewer DwFR,
>
> Thank you again for your valuable feedback and suggestions. In our previous response, we have provided a detailed explanation and supplemented our arguments with theoretical analysis and experimental validation. To help you better understand the improvements we made, I would like to summarize the key points:
>
> 1. **One of our main contributions:** Our method introduces per-action-dimension correction terms that can be exactly and efficiently calculated by TPMs to enhance the action sampling procedure. These correction terms significantly improve the performance, and our method does not necessarily need to be combined with rejection sampling or beam search to be effective. For example, **DT-based Trifle achieves significant performance gains compared to DT without using any rejection sampling**, proving that the per-action-dimension correction terms play a critical role in guiding the action generation process.
>
> 2. **Additional ablation study over rejection sampling:** We provided additional ablation studies (**Table 1** in the attached PDF) to show that
>
>     (i) The proposed correction term works well across different base algorithms (DT and TT). Specifically, as indicated by the last row of Table 1(b) in tha attached PDF, TT w/o rejection sampling suffers a significant performance drop, while **Trifle works well even w/o rejection sampling (competitive results as TT w/ beam search)**.
>
>     (ii) Trifle can be effectively combined with rejection sampling or beam search to further boost performance, which leads to state-of-the-art results in 7 out of 9 Gym-MuJoCo environments.
>
> 3. **Another equally important contribution:** Our method enables **exact and efficient multi-step value estimation**, which significantly enhances performance in stochastic environments as shown in Section 6.2.
>
> We believe we have addressed the concerns raised, but if you have any further questions or issues, we would be happy to discuss them with you. We look forward to your feedback.
>
> Thank you again for your time and consideration.

---

### Official Review · Reviewer_jZtp · 2024-07-30

**Soundness:** 3
**Presentation:** 4
**Contribution:** 3
**Rating:** 7
**Confidence:** 4

**Summary:**

The paper introduces Trifle (Tractable Inference for Offline RL) that leverages Tractable Probabilistic Models (TPMs) to enhance the performance of offline RL tasks. The paper emphasizes that beyond the expressiveness of sequence models, tractability--efficiently answering probabilistic queries--is crucial for accurate evaluation in offline RL, particularly in environments with high stochasticity and action constraints. In particular, Trifle uses a type of TPMs called Probabilistic Circuits (PCs), which supports the computation of arbitrary marginal and conditional probabilities in linear time with respect to their size. In practice, Trifle uses a mixture of single-step (return-to-go) and multi-step value estimates to condition action generation, allowing Trifle to handle both optimal and suboptimal return-to-gos effectively. Trifle enhances traditional rejection sampling by incorporating TPMs to adjust the proposal distribution, improving the likelihood of sampling high-return actions. It employs beam search to maintain and refine high-probability action sequences, ensuring actions are within the offline data distribution and likely to yield high rewards. Additionally, Trifle uses an adaptive thresholding mechanism to dynamically select expected return thresholds, maintaining robust performance across various evaluation settings.

The paper provides comprehensive empirical comparisons across nine Gym-MuJoCo benchmarks, a stochastic Taxi environment, and action-space-constrained tasks. Trifle consistently outperforms state-of-the-art baselines, including Trajectory Transformers (TT), Decision Transformers (DT), and various imitation learning and offline TD learning methods. These results demonstrate Trifle's robustness and effectiveness, particularly in challenging stochastic and constrained environments, underscoring its potential to advance offline RL methodologies.

**Strengths:**

**Originality**: The paper presents a novel perspective by highlighting the importance of tractability in offline RL. By introducing TPMs into this domain, it addresses a gap not thoroughly explored by existing approaches, which typically focus on the expressiveness of sequence models.

**Quality**: The methodology is robust and well-supported by theoretical insights and extensive empirical evaluations. The experiments demonstrate that Trifle significantly outperforms strong baselines on diverse benchmarks, including challenging stochastic environments and safe RL tasks with action constraints.

**Clarity**: The paper is well-written and organized, making complex concepts accessible. The inclusion of detailed experimental setups and comprehensive results aids in understanding the advantages of Trifle over existing methods. I appreciate the overview on TPMs and the build up from section 3 to 5 from the theoretical to the practical implementation.

**Significance**: By demonstrating that tractability can greatly enhance evaluation-time performance, the paper paves the way for future research to develop more inference-aware RL approaches. The results on diverse and challenging benchmarks highlight the practical benefits of integrating TPMs into RL algorithms.

**Weaknesses:**

1. Limited evaluations on complex tasks (e.g. in the D4RL benchmark beyond the nine studied). There is perhaps a challenge (computational?) to scale up the PCs on more complicated environments. Even if the authors do not provide the results for these, perhaps it would be worth discussing any challenges on scaling up Trifle to these more complex tasks.
2. More detailed ablation studies. The paper includes some ablation studies, such as comparisons of Trifle variants with and without Q-value-based action filtering, demonstrating the effectiveness of exact inference. However, it would benefit from additional detailed ablations to isolate the contributions of other components, such as the adaptive thresholding mechanism and beam search strategy, to offer deeper insights into which elements are most critical to Trifle's performance improvements.
3. Scalability / Efficiency Analysis. The paper lacks a thorough analysis of Trifle's computational complexity and scalability. Discussing the trade-offs between the benefits of tractability and the computational overhead introduced, especially in high-dimensional action spaces, would provide a clearer picture of Trifle's practical feasibility. Including detailed runtime comparisons with other methods would be beneficial. The current runtime analysis indicates a 1.5 to 3 times increase in inference time compared to base models, but a more comprehensive analysis would be useful (e.g. plotting out a scaling law curve for Trifle vs TT in table 5 with more horizon, etc.)

**Questions:**

Related to the above’s weaknesses:
1. How does the adaptive thresholding mechanism and beam search hyperparameters affect the performance of Trifle across different environments and datasets? Having some ablation studies on this would be helpful to suggest practical values for future work.
2. Could you provide more intuitive explanations and discuss the practical implications of the theoretical guarantees provided in Sections 4.1 and 4.2? While the theoretical results are well-presented, further clarity on how these results guide the design and implementation of Trifle would help readers better understand the theoretical contributions and their impact on performance. Specifically, discussing practical scenarios where these theoretical guarantees are most beneficial would strengthen the paper.
3. Have you considered evaluating Trifle on more complex tasks like the Antmaze benchmark?

Post rebuttal: increased my score from 6 to 7.

**Limitations:**

The authors have addressed some limitations of their work, specifically the dependency on expressive TPMs and the current inefficiency of PCs compared to neural network packages.  However, there is no discussion on the potential negative societal impact of their work as the method was tested on simulated environments with no immediate societal impact.

One possible limitation to discuss is also how Trifle scales in high-dimensional action spaces, which might introduce computational overhead and impact performance.

---

> ### Author Rebuttal · Authors · 2024-08-07
>
> ### Comment #1: practical implications of the theoretical guarantees in Secs. 4.1 and 4.2
>
> Thanks for the constructive comment. The theoretical results are used to elaborate two key inference-side challenges. Both challenges are introduced in the general comment, and we provide further details in the following. We will incorporate the following discussion into the next version of the paper.
>
> **Sampling actions conditioned on high expected return**
>
> As described in Section 3, our goal is to sample from $p(a_t | s_{0:t}, \mathbb{E}[V_{t}] \geq v)$. However, Theorem 1 illustrates that it is NP-hard to exact compute even when $p(s_{0:t}, a_{t}, V_{t})$ follows a simple Naive Bayes distribution, which guides Trifle to find *good approximations*.
>
> Specifically, we utilize TPM to compute the correction term $p_{TPM} (V_t \geq v | s_t, a_{t}^{\leq i})$ to bias the actions generated by any prior policy towards high expected return $\mathbb{E}[V_t]$. Note that we compute the correction term exactly for each dimension $a_t^i$, which is highly non-trivial as we have to marginalize out unseen action variables $a_t^{i+1},...,a_t^k$ and this forms an exponentially large action space. We have **more ablation studies** on the effect of the TPM-provided terms in the response to comment #2.
>
> **Estimating state-action values under stochastic transition dynamics**
>
> As discussed in the general comment, we need to use multi-step value estimates under stochastic environments for better accuracy. While it is generally hard to obtain low-variance estimates, with TPMs' ability to compute arbitrary conditional distributions, we can efficiently compute the multi-step value estimate in Eq. (3). This allows us to enjoy the better accuracy of the multi-step value estimates while mitigating the inefficiency to compute them.
>
> ### Comment #2: effect of the beam search hyperparameters
>
> We conduct additional ablation studies on both the adaptive thresholding mechanism and rejection sampling/beam search.
>
> **Ablation studies on the adaptive thresholding mechanism**
>
> We report the performance of TT-based Trifle with variant $\epsilon$ vs TT on Halfcheetah Med-Replay in **Table 1(a)** of the rebuttal PDF. Trifle is robust to $\epsilon$ and consistently outperforms TT.
>
> We also conduct ablation studies comparing the performance of the adaptive thresholding mechanism with the **fixed thresholding mechanism** on two environments in **Table 1(b)** by fixing different $v$s. The table shows that the adaptive approach consistently outperforms the fixed value threshold in both environments.
>
> **Ablation studies on the rejection sampling/beam search**
>
> As mentioned in the general response, for the Gym-MuJoCo benchmark, we only adopt rejection sampling/beam search for TT-based Trifle and not for DT-based Trifle. Therefore, the success of DT-based Trifle strongly justifies the effectiveness of the TPM components.
>
> For TT-based Trifle, we adopted the same beam search hyperparameters as reported in the TT paper. We conduct ablation studies on beam search hyperparameters in **Table 2** of PDF to investigate the effectiveness of Trifle's each component:
>
> -  Trifle consistently outperforms TT across all beam search hyperparameters and is more robust to variations of both planning horizon $H$ and beam width $W$.
> - (a) Trifle w/ naive rejection sampling >> TT w/ naive rejection sampling (b) Trifle w/o rejection sampling >> TT w/o rejection sampling. In both cases, Trifle can positively guide action generation.
> - Trifle w/ beam search > Trifle w/ naive rejection sampling > Trifle w/o rejection sampling >> TT w/ naive rejection sampling. Although other design choices like rejection sampling/beam search help to better approximate samples from the high-expected-return-conditioned action distribution, the per-dimension correction terms computed by Trifle play a very significant role.
>
> ### Comment #3: computational complexity analysis
>
> Thanks for the valuable suggestion.
>
> First, we conduct a more detailed runtime analysis, and the main results are shown in Figure 1 of the attached PDF.
>
> Figure 1 (left) in the rebuttal PDF expands Table 5 and plots the step-wise inference-time scaling curve of Trifle vs TT with varying horizons. As we increase the horizon, the relative slowdown is mitigated. This is because, across different horizons, TPM-related computation consistently requires ~1.45s computation time, which additional computation overhead is diminishing as we increase the beam horizon.
>
> Moreover, Trifle is efficient in training. It only takes 30-60 minutes to train a PC (the adopted TPM) on one GPU for each Gym-Mujuco task (note that we only need one PC per task). In comparison, training the GPT model for TT takes approximately 6-12 hours (80 epochs).
>
> Note that there are recent breakthroughs [1] on designing efficient PC implementations, which can significantly speed up training and inference of Trifle.
>
> [1] Liu et. al. "Scaling Tractable Probabilistic Circuits: A Systems Perspective." arXiv preprint arXiv:2406.00766 (2024).
>
> ### Comment #4: evaluation on the Antmaze task
>
> We tried to run the Antmaze task but have not successfully reproduce the performance of TT+Q (as reported by the original paper, the vanilla TT fails due to the sparsity of the rewards) since there is no official implementation on GitHub and we have not found third-party implementation for this. We will keep trying to reproduce during the discussion period.
>
> ### Comment #5: potential negative societal impact
>
> Thanks. We will include the following discussion:
>
> This paper proposes a new offline RL algorithm, which aims to produce policies that achieve high expected returns given a pre-collected dataset of trajectories generated by some unknown policies. When there are malicious trajectories in the dataset, our method could potentially learn to mimic such behavior. Therefore, we should only train the proposed agent in verified and trusted offline datasets.

---

> ### Author Response · Authors · 2024-08-07
> **Details of the Adaptive Thresholding Mechanism**
>
> the adaptive thresholding mechanism is adopted when computing the term $p_{TPM} (V_t \geq v | s_t, a_{t}^{\leq i})$ of Equation (2), where $i \in \{1, \dots, k\}$, $k$ is the number of action variables and $a_t^{i}$ is the $i$th variable of $a_t$. Instead of using a fixed threshold $v$, we choose $v$ to be the $\epsilon$-quantile value of the distribution $p_{TPM}(V_t|s_t,a_{t}^{< i})$ computed by the TPM, which leverage the TPM's ability to exactly compute marginals given incomplete actions (marginalizing out $a_{t}^{i:k}$). Specifically, we compute $v$ using $v = max_r\{r\in\mathbb{R}|p_{TPM}(V_t\geq r|s_t,a_{t}^{< i})\geq 1-\epsilon\}$. Empirically we fixed $\epsilon$ for each Gym-MuJoCo environment and $\epsilon = 0.2$ or $0.25$, which is selected by runing grid search on $\epsilon \in [0.1,0.25]$.

---

> ### Author Response · Authors · 2024-08-07
> **Discussion about Trifle's Scalability**
>
> The scalability of Trifle to more complex tasks is also an interesting question. Some of the key challenges to scaling up an offline RL algorithm include (i) high-dimensional action spaces, and (ii) more complex transition dynamics and policy. Figure 1 (right) shows that Trifle's runtime (TPM-related) scales linearly w.r.t. the number of action variables, which indicates its efficiency for handling high-dimensional action spaces. The second largely depends on the performance of modeling complex sequences. While we do not have a definite answer on the scalability of the adopted TPMs, there are abundant recent papers that significantly improve the expressive power of TPMs.

---

> > ### Comment · Reviewer_jZtp · 2024-08-14
> >
> > Thank you for the detailed response to my concerns and questions, along with the additional results in the rebuttal PDF. This helped clarify the implications of the theorems, the effects of the beam search parameters, the computational complexity / scalability, and details of the adaptive thresholding mechanism. I encourage the authors to incorporate these details in the final version of the paper. I am raising my score from 6 to 7.

---

> ### Author Response · Authors · 2024-08-14
>
> Thank you very much for supporting our paper and increasing the score! And thanks for your valuable suggestions during the rebuttal period to help us improve the paper. We will incorporate the additional details in the final version according to your comments. We sincerely appreciate your feedback!

---

### Author Rebuttal · Authors · 2024-08-07

We thank all reviewers for their constructive feedback and for acknowledging our paper as novel, well-presented, and comprehensively evaluated. We summarize common questions and concerns raised by the reviewers in the following.

**Key technical differences of Trifle compared to other RvS or offline RL algorithms**

As highlighted in our paper as well as in the literature (e.g., [1]), there are several challenges in RvS algorithms from a tractable inference perspective. This paper identifies two such challenges and proposes to solve them using **tractable probabilistic models**.

**Challenge #1**: Sampling actions condition on high **expected** return

Given a generative model of the joint probability distribution $p(s_{0:t}, a_{0:t}, V_{0:t})$, our goal is to sample actions $a_t$ condition on the current state $s_{0:t}$ and high expected future returns (i.e., high $\mathbb{E}[V_t]$), which can be concisely written as $p(a_t | s_{0:t}, \mathbb{E}[V_t] \geq v)$ (Eq. (1) in the paper). However, the condition on the expected future return makes it a hard inference task given generative models over the joint distribution. We show that this query is NP-hard to compute exactly even when the joint distribution follows Naive Bayes (Thm. 1).  We propose a promising approximate sampling algorithm for $p(a_t | s_{0:t}, \mathbb{E}[V_t] \geq v)$ that leverages the ability of TPMs to compute arbitrary conditional distributions. This component can be directly used to replace policy components such as $p(a_t | s_{0:t})$ (e.g., TT) and $p(a_t | s_{0:t}, V_t = v)$ (e.g., DT) used in existing RvS algorithms. Empirical evaluations demonstrate the addition of this component significantly improves over the base algorithm.

**Challenge #2**: Estimating state-action values under stochastic transition dynamics

Another major challenge comes from the inaccuracy of the RTG), an estimate of the return, provided in the offline datasets. This problem is even worse in stochastic environments, where the labeled RTGs have very high variation. To combat this inaccuracy, it is common to use multi-step value estimates such as TD(1) and TD(λ). Specifically, we want to compute $\mathbb{E} [r_t + \cdots + r_{t'} + V_{t'+1} | s_{0:t}, a_{t:t'}]$ give a joint distribution represented by a generative model. However, computing such values requires implicitly marginalizing out future states $s_{t+1:t'}$, which is computationally intractable for most models. We propose to use TPM to exactly compute such terms, which leads to a significant performance gain in stochastic environments.

**The relationship between Trifle and rejection sampling/beam search algorithm**

The key insight of Trifle to solve challenge #1 is to utilize tractable probabilistic models to better approximate action samples from the desired distribution $p(a_t | s_{0:t}, \mathbb{E}[V_t] \geq v)$. We highlight that the most crucial design choice of our method for this goal is that: Trifle can effectively bias the per-action-dimension generation process of any base policy towards high expected returns, which is achieved by adding per-dimension correction terms $p_{TPM} (V_t \geq v | s_t, a_{t}^{\leq i})$ (Eq. (2) in the paper) to the base policy.

While the rejection sampling method can help us obtain more unbiased action samples through a post value(expected return)-estimation session, we only implement this component for TT-based Trifle (not for DT-based Trifle) for fair comparison, as the DT baseline doesn't perform explicit value estimation or adopt any rejection sampling methods. Moreover, the beam search algorithm also comes from TT. Although it is a more effective way to do rejection sampling, it is not the necessary component of Trifle, either.

Next, following the suggestions of the reviewers, we conduct a comprehensive ablation study over beam search/rejection sampling in Table 2 of the attached PDF. The results:
- **Trifle with beam search > Trifle with naive rejection sampling > Trifle w/o rejection sampling >> Base policy w/o rejection sampling**
- **Trifle w/o rejection sampling >> TT with naive rejection sampling**
- **Trifle with naive rejection sampling $\approx$ TT with beam search**

strongly justifies our claim. (We analyze the results in detail to each reviewer)

**Computational efficiency/scalability of Trifle**

For the inference-side efficiency, we conduct a more detailed inference-time evaluation, and the main results are shown in Figure 1 of the attached PDF. Specifically, Figure 1 (left) expands Table 5 and plots an inference-time scaling curve of Trifle vs TT with varying horizons, where we can draw the same conclusion as Appendix D.3 of the paper: Since TPM-related computation consistently requires ~1.45s computation time across different horizons, the relative slowdown of Trifle is diminishing as we increase the beam horizon. Figure 1 (right) shows that Trifle's runtime (TPM-related) scales **linearly** w.r.t. the number of action variables, which indicates its efficiency for handling high-dimensional action spaces.

Trifle is also efficient in training. It only takes 30-60 minutes (~20s per epoch, 100-200 epochs) to train a PC on one GPU for each Gym-Mujuco task (**Note that a single PC model can be used to answer all conditional queries required by Trifle**). In comparison, training the GPT model for TT takes approximately 6-12 hours (80 epochs).

Note that there are recent breakthroughs [2] in designing efficient PC implementations, which can **significantly speed up** the computation of Trifle (both training and inference).

[1] Paster, Keiran, Sheila McIlraith, and Jimmy Ba. "You can’t count on luck: Why decision transformers and rvs fail in stochastic environments."

[2] Liu, Anji, Kareem Ahmed, and Guy Van den Broeck. "Scaling Tractable Probabilistic Circuits: A Systems Perspective."

---

### Decision · Program_Chairs · 2024-09-25

**Decision:**

Accept (poster)

**Comment:**

The reviewers generally liked the paper and has unanimous positive reviews so we are recommending acceptance of this paper. That said, we would like to point out that it is unclear if the paper has sufficient comparisons to demonstrate the utility of the framework, for instance, results on the more challenging antmaze tasks are absent, on other gym environments the performance gains are tiny, so this is largely a simple framework. That said, the didactic experiments are nice. I would recommend acceptance conditioned on adding a more extensive set of results for the final version.